# MAIVeSS: streamlined selection of antigenically matched, high-yield viruses for seasonal influenza vaccine production

Cheng Gao[1,2,3,4,14], Feng Wen[5,14], Minhui Guan[1,3,4], Bijaya Hatuwal[1,2,3,4], Lei Li [6,7], Beatriz Praena[1,3,4], Cynthia Y. Tang [1,3,4,8], Jieze Zhang [9], Feng Luo [10], Hang Xie [11], Richard Webby [12], Yizhi Jane Tao[13] & Xiu-Feng Wan [1,2,3,4,5,8] ✉

Vaccines are the main pharmaceutical intervention used against the global public health threat posed by influenza viruses. Timely selection of optimal seed viruses with matched antigenicity between vaccine antigen and circulating viruses and with high yield underscore vaccine efficacy and supply, respectively. Current methods for selecting influenza seed vaccines are labor intensive and time-consuming. Here, we report the Machine-learning Assisted Influenza VaccinE Strain Selection framework, MAIVeSS, that enables streamlined selection of naturally circulating, antigenically matched, and high-yield influenza vaccine strains directly from clinical samples by using molecular signatures of antigenicity and yield to support optimal candidate vaccine virus selection. We apply our framework on publicly available sequences to select A(H1N1)pdm09 vaccine candidates and experimentally confirm that these candidates have optimal antigenicity and growth in cells and eggs. Our framework can potentially reduce the optimal vaccine candidate selection time from months to days and thus facilitate timely supply of seasonal vaccines.

Vaccination is a major strategy for preventing influenza virus infections. However, hemagglutinin (HA), the primary target of these vaccines, can undergo antigenic change, enabling virus to evade existing host immunity elicited by natural infections and/or vaccinations. Annual updates of vaccine composition are conducted to match between vaccine and circulating viruses. This is a resource and time-consuming process that requires global private and public collaboration coordinated through the World Health Organization (WHO) Global Influenza Surveillance and Response System (GISRS)[1,2].

A major activity of GISRS is to make global recommendations for the most appropriate vaccine viruses and to provide corresponding candidate vaccine viruses (CVVs) for distribution to manufacturers of live attenuated and inactivated vaccines. Once sourced, manufacturers then further optimize CVV growth properties for use in vaccine manufacture. An ideal CVV must have the appropriate antigenic properties,

[1]Center for Influenza and Emerging Infectious Diseases, University of Missouri, Columbia, MO 65211, USA. [2]Department of Electrical Engineering & Computer Science, College of Engineering, University of Missouri, Columbia, MO 65211, USA. [3]Department of Molecular Microbiology and Immunology, School of Medicine, University of Missouri, Columbia, MO 65211, USA. [4]Bond Life Sciences Center, University of Missouri, Columbia, MO 65211, USA. [5]Department of Basic Sciences, College of Veterinary Medicine, Mississippi State University, Starkville, MS 39762, USA. [6]Department of Chemistry, Georgia State University, Atlanta, GA 30303, USA. [7]Center for Diagnostics & Therapeutics, Georgia State University, Atlanta, GA 30303, USA. [8]Institute for Data Science and Informatics, University of Missouri, Columbia, MO 65211, USA. [9]Department of Bioengineering, Rice University, Houston, TX 77030, USA. [10]University School of Computing, Clemson University, Clemson, SC 29634, USA. [11]Laboratory of Respiratory Viral Diseases, Division of Viral Products, Office of Vaccines Research and Review, Center for Biologics Evaluation and Research, US Food and Drug Administration, Silver Spring, MD 20993, USA. [12]Department of Infectious Diseases, St. Jude Children's Research Hospital, Memphis, TN 63141, USA. [13]Department of BioSciences, Rice University, Houston, TX 77251, USA. [14]These authors contributed equally: Cheng Gao, Feng Wen. ✉e-mail: wanx@missouri.edu

maintain these properties through the production pipeline, and have a high growth phenotype. Timely selection of an effective high-yielding CVV is critical for optimal seasonal influenza vaccine manufacturing. In the 2003–2004 influenza season, an A/Fujian/411/2002-like virus was preferred as the recommended A(H3N2) vaccine virus, but the inability to identify CVVs led to a vaccine mismatch[3]. During the A(H1N1)pdm09 pandemic, vaccine supply was delayed due to the poor yield of initial CVVs[4], and a global vaccine campaign was not initiated until better yielding viruses were obtained which was after the second pandemic wave[5].

Efforts to generate high-yield CVVs for vaccine manufacturing in embryonated chicken eggs or cultured cells continue year-round even before recommended vaccine viruses are finalized for the upcoming influenza season. The conventional strategy to achieve high yield often involves additional passages in eggs or cells[6], and genetic approaches rely on reassortment with a donor strain that exhibits high yield traits in eggs or cells[7]. Both approaches may take up to 6 months as well as have limitations. Egg or cell adaptation can result in undesired antigenic changes due to additional mutations in HA and reassortment strategies do not always lead to substantial improvements in yield[8–11]. Therefore, identifying influenza vaccine viruses with high-yield phenotypes directly from sequences obtained from clinical samples would be ideal and could potentially accelerate vaccine production timelines.

Over the past few years, several computational models[12–14] have been developed to identify influenza antigenic variants using genomic sequences (See the discussion section in Supplementary Information (SI)). However, none of these models can be used to directly identify antigenic match and high-yield viruses based on genetic sequences.

In our study, we propose an approach to overcome challenges in influenza vaccine strain selection. We introduce MAIVeSS, a framework that employs machine learning algorithms, to predict antigenic and yield phenotypes using viral genomic sequences from clinical samples. We validated MAIVeSS by screening A(H1N1)pdm09 for ideal CVVs and confirmed the high yield nature of the identified CVVs in both cells and eggs. Our results show that that MAIVeSS can facilitate the selection of naturally circulating influenza vaccine strains with matching antigenicity and high-yield as seed viruses for influenza vaccine production.

## Results

### Machine-learning assisted influenza vaccine strain selection framework

This study aimed to design MAIVeSS to learn genetic features associated with three key influenza virus biological properties: antigenicity, growth, and receptor-binding (Fig. 1). We implemented and compared a set of machine learning models within MAIVeSS and found that the multi-task learning group-guided sparse learning model (MTL-GGSL) outperformed other state-of-the-art models for predicting antigenicity and glycan binding, while the generalized hierarchical sparse model (GHSM) outperformed other models for assessing growth (see Supplementary Data 1–3).

Using the features learned, MAIVeSS scores CVVs using a query HA protein sequence based on two properties: (1) antigenic properties, and (2) yield properties in eggs and/or cells ($HY^{cell}$, high-yield in cells;

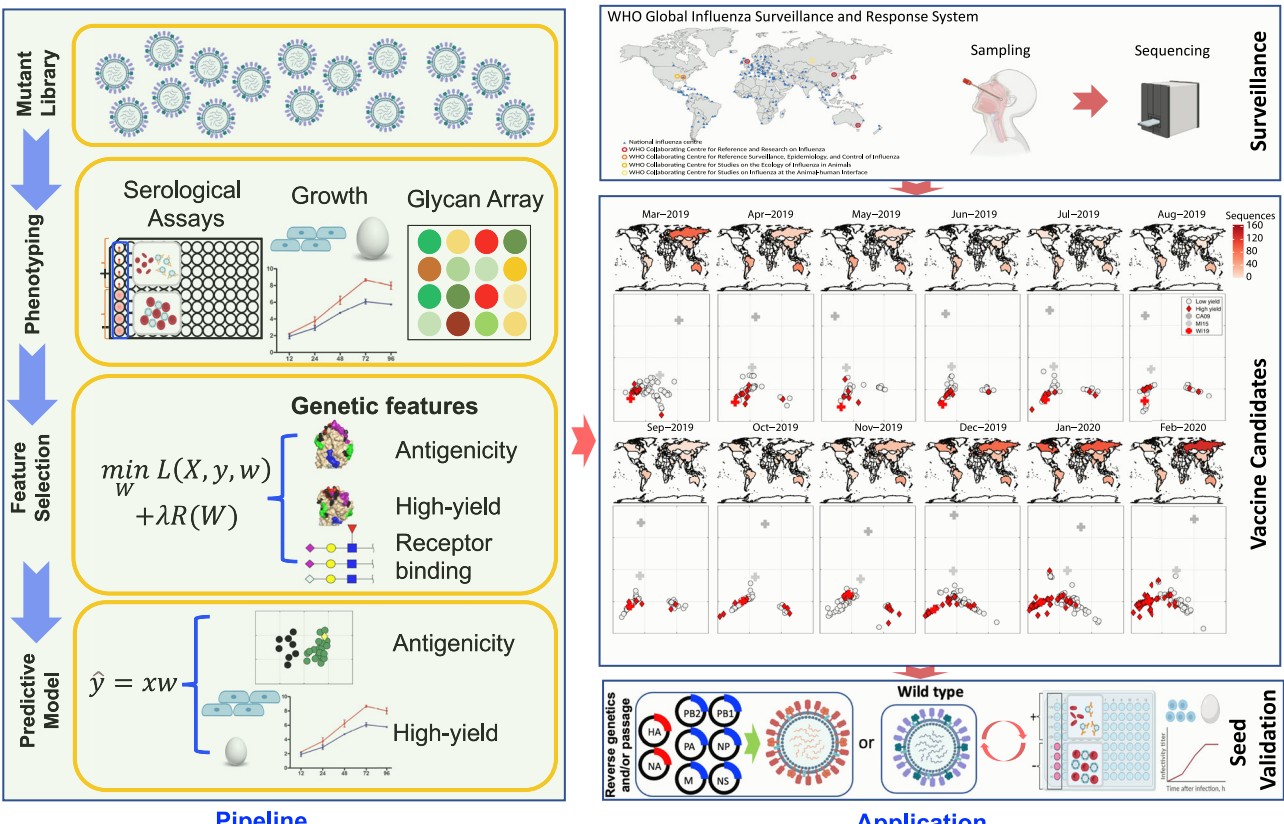

**Fig. 1 | Machine-learning Assisted Influenza VaccinE Strain Selection framework (MAIVeSS).** This model has been developed to select high-yield vaccine seeds that match the antigenicity of circulating influenza strains. To achieve this, a library of viruses with mutations in the receptor binding sites of the HA protein is generated and analyzed for their serological reactivity, replication efficiency in eggs and cells, and glycan profiling using microarrays. A sparse learning model will then be applied to identify genetic features that correlate with these phenotypes. Based on these features, a predictive model will be developed to quantify antigenic distances from the currently used vaccine strains and the yield ability in eggs and cells using HA protein sequences as input data. This strategy can be adapted for other subtypes of influenza viruses, and the model can also be modified to incorporate NA sequences. Part of this figure was created with BioRender.com.

HY[egg], high-yield in eggs; HY[both], high-yield in both cells and eggs; LY[cell], low-yield in cells; LY[egg], low-yield in eggs; LY[both], low-yield in both cells and eggs). In this study, the WT virus is defined as a reassortant with wild-type HA and NA genes from A/California/04/2009(H1N1pdm09) (CA/04) and six internal genes from A/Puerto Rico/8/1934(H1N1) (PR8). High-yield is defined as a >10-fold increase in $TCID_{50}$/mL compared to the WT on the same substrate. By leveraging these predictive models, MAIVeSS can rapidly identify influenza viruses that are both antigenically matched and high-yielding from genome sequences obtained during surveillance. MAIVeSS is accessible through both GitHub (https://github.com/FluSysBio/MAIVeSS) and a webserver (http://sysbio.missouri.edu/software/MAIVeSS).

In this study, we demonstrated the effectiveness of our machine learning models using A(H1N1)pdm09 viruses as an exemplary application, but the same principles can be readily applied to other subtypes of influenza viruses.

## Development of an A(H1N1)pdm09 mutant library for machine learning

To enhance the reliability of feature selection for high-yield viruses, we established a random mutant virus library that targets the HA receptor binding site (RBS) of CA/04[15]. All the mutants were subjected to antigenic analyses using hemagglutination inhibition (HAI) assays, yield analyses in both MDCK cells and embryonated chicken eggs, and receptor-binding profiling through glycan microarrays. The phenotypic data collected were then used as training and testing data in MAIVeSS to identify the molecular features associated with antigenicity and yield and to establish predictive models.

In total, we generated 822 HA-containing plasmids, each carrying one to seven random mutations within or near the HA RBS (residues 119–241, H1 numbering; 126–244, H3 numbering). Using these mutant plasmids, we then generated corresponding mutant viruses via reverse genetics. Rescued mutant viruses had an HA from the mutant pool, a NA gene from CA/04, and the remaining 6 gene segments from PR8. To increase the likelihood of successful virus rescue, the transfection products underwent three passages. If none of the three passages yield positive results, we will conclude that the mutant cannot be rescued.

As a result, a total of 189 mutant viruses bearing unique amino acid substitutions with different biochemical properties were successfully generated (Supplementary Fig. 1). Of the mutant viruses that were successfully rescued, 96 had substitutions within the 119–190 region of the HA, 15 had substitutions within the 190–241 region, and 78 had substitutions in both regions. The positions of the substitutions overlapped with the reported HA antigenic sites Sa ($n = 11$), Sb ($n = 9$), Ca1 ($n = 7$), and Ca2 ($n = 7$). For consistency, specific amino acid positions in the mutants described in the following context are numbered according to the H3 numbering system. Eighty of the mutant viruses had a single substitution, 80 had two, and 29 had three or more. Interestingly, all substitutions present in the rescued viruses were located outside of the receptor binding site (RBS) and did not directly interact with the receptor molecule (Fig. 2A). In contrast, virus rescue was unsuccessful when substitutions were present within the RBS pocket.

## Most mutations did not alter antigenic properties

To determine the antigenic properties of the mutant viruses generated, we performed HAI assay using post-infection ferret antisera. Out of 189 mutant viruses, only 5 mutants had a ≥4-fold reduction in their HAI titers relative to the antisera's homologous virus (Supplementary Data 4). These 5 antigenically distinct mutants had at least one substitution in known HA antigenic sites, with other substitutions mostly present within or close to the Ca1, Ca2, Sa, or Sb regions[16–19]. Of note, ferret antisera generated against WT CA/04 were unable to neutralize the triple mutant HA D131E-S193T-A198S.

We integrated the serological data of the 189 mutants with archived public data for seasonal A(H1N1) (1977-2009) and A(H1N1) pdm09 viruses (2009–2016)[20]. By using these integrated HA sequence and serological data, we utilized MAIVeSS to identify residues that were associated with antigenic changes. The results showed that 30 residues were associated with the antigenicity of A(H1N1) viruses (Table 1 and Supplementary Data 5), and most of these residues were located within or close to the antigenic sites, particularly Ca1, Ca2, Sa, and Sb (Fig. 2B). Of these 30 residues, only position 225 has been reported in the literature to potentially harbor an egg-adapted substitution for 2009 H1N1 viruses (Table 1 and Supplementary Information).

## Amino acid substitutions near the HA RBS can result in high-yield traits in both cells and eggs

We next assessed how the amino acid substitutions introduced affected virus yield in both cells and eggs by measuring the infectious titers ($TCID_{50}$) for each mutant after growth. We identified 14 HY[cell] mutants that showed at least a 10-fold increase in virus yield compared to the WT virus as well as 29 LY[cell] mutants that showed at least a 10-fold decrease (Supplementary Fig. 2 and Supplementary Data 4). The highest yield was observed in the HA N159D-K166I mutant, with a yield of $1.52 \times 10^7$ $TCID_{50}$/mL, which was about 100-fold higher than WT. Additionally, 33 HY[egg] mutants and 19 LY[egg] mutants were identified. The HA D131E-S193T-A198S, HA N159D-K166I, and HA I169F-D225G mutants had the highest titers in eggs, which were approximately 800-fold higher than the WT virus. Of note, these three mutants also exhibited high-yield traits in cells and were thus designated as HY[both].

By using the HA protein sequence and their associated yield data, we applied MAIVeSS and identified 38 residues were associated with virus yield (Table 2 and Supplementary Data 6). The majority of these residues were located on the surface of the HA trimer and in close proximity to the RBS pocket (Fig. 2C). Interestingly, we observed that different substitutions at the same position could lead to different outcomes. For example, a change from small polar amino acids to nonpolar amino acids at residue 142 was predicted to enhance virus yield, whereas substitution to polar/charged amino acids at the same position was predicted to reduce virus yield in eggs and cells (Table 2).

## Diversified glycan binding facilitates virus replication in cells and eggs

To investigate if the high-yield trait correlates with glycan substructure binding properties, we analyzed the receptor-binding properties of the 189 mutant viruses using glycan microarrays comprising of 75 glycoforms (Supplementary Fig. 3). The binding signals to these glycan isoforms varied widely among the mutants (Supplementary Fig. 4). Notably, all mutants exhibited strong binding avidity to glycans that were terminated with SA2,6 Gal, as we had expected.

We further used a matrix of 27 glycan substructure features to group the glycans based on their internal and terminal substructures as well as their linkers (Supplementary Fig. 5). Our analysis revealed that HY[cell] mutants displayed elevated binding avidities to glycans terminated with Neu5Acα2-6Galβ1-4GlcNAc (6'SLN), but not to Neu5Acα2-3Galβ1-4GlcNAc (3'SLN), Neu5Aca2-3Galβ1-4(Fuca1-3) GlcNAc (sLe[X]) or Neu5Gcα2-3Galβ1-4GlcNAc. In contrast, HY[egg] and HY[both] mutants (such as HA D131E-S193T-A198S and HA I169F-D225G) showed increased binding affinities to glycans terminated with 3'SLN and sLe[X]. Interestingly, a few HY[egg] and HY[both] mutants also exhibited significant increases in binding avidities to a glycan that is terminated with Neu5Gcα2-6Galβ1-4GlcNAc or Neu5Gcα2-3Galβ1-4GlcNAc.

By employing biolayer interferometry analyses (BLI) for glycan binding profiling, we confirmed the broadened binding specificity of the HY[both] mutant HA D131E-S193T-A198S. Specifically, we

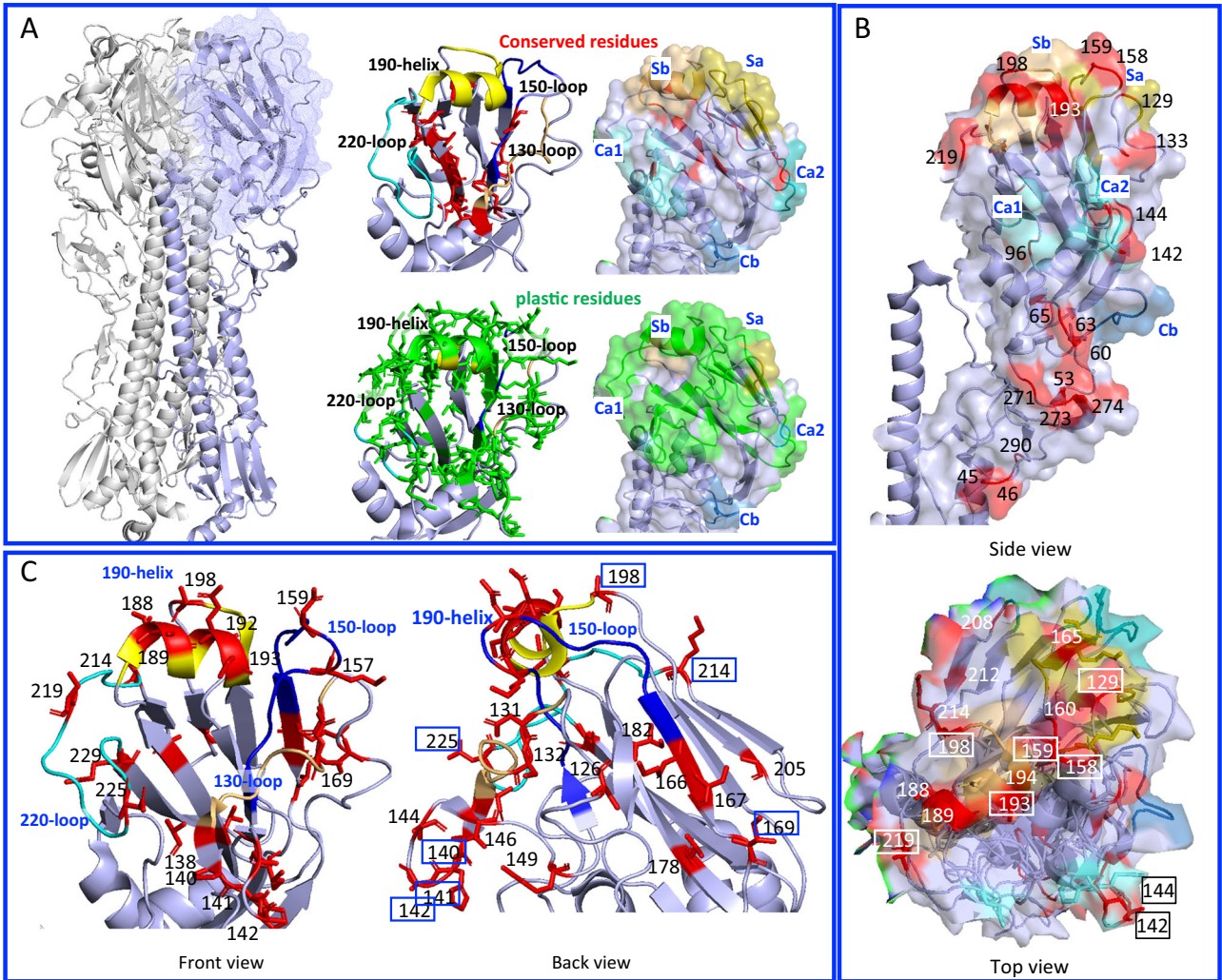

**Fig. 2 | Characterization of mutant viruses targeting the receptor-binding site of the HA protein.** The HA of A/California/04/2009(H1N1) (CA/04) was used as the template to develop the error-prone PCR mutant library. **A** The regions in the HA receptor-binding site where viable mutant viruses were obtained are shown in green, while the regions where no viable mutant viruses were obtained are shown in red. The receptor binding pocket includes the 130-loop, 150-loop, 190-helix, and 220-loop. **B** The key residues associated with antigenicity are located in five HA antigenic sites: Sa (in purple), Sb (in raspberry), Ca1/Ca2 (in green), and Cb (in blue), as well as outside these antigenic sites (in orange). **C** The key residues associated with virus yield in cells and eggs are located near the receptor binding site or outside the receptor binding site. The residues are shown in two views of the structures, and the overlapped residues are boxed. The HA protein structure was modeled using PyMOL with the CA/04HA protein structure from the Protein Databank (PDB) 3LZG as the template.

demonstrated that this mutant not only binds to 6'SLN, but also to 3'SLN and sLeX (Supplementary Fig. 6).

We used MAIVeSS (see Online Methods) to identify the glycan substructures associated with yield traits in cells and eggs associated with binding preference to these glycan substructures (Supplementary Data 7–11). Our analysis revealed several glycan terminal substructures that were significantly associated with high-yield traits, including 6'SLN, 3'SLN, sLe[x], and Neu5Gcα2-6Galβ1-4GlcNAc. Additionally, we found that certain internal substructures, such as core lactose, GlcNAcb1-2, and Galα1-4Galβ1-4GlcNAc, had a significant impact on glycan binding.

## A subset of antigenically matched A(H1N1)pdm09 epidemic viruses were high-yield in both cells and eggs

We utilized MAIVeSS to predict both yield in eggs and cells and antigenic properties of sequenced A(H1N1)pdm09 viruses (2009−2020, n = 11,424) (Supplementary Data 12). Using the antigenic distance matrix generated by MAIVeSS, we created a sequence-based antigenic cartography map, which revealed two antigenic clusters, CA/09 and WI/19 (Fig. 3). The acquisition of N159K in the Sa antigenic site was predicted to cause an antigenic drift of the A/Wisconsin/588/2019(H1N1) (WI/19) HA from that of CA/09, which was consistent with those reported in other studies[21,22].

Using MAIVeSS as the prediction tool, a total of 155 viruses were identified as potential HY[egg], 433 as HY[cell], and 761 as HY[both]. Of the 1349 viruses identified as high-yield variants, 897 had HA sequences that were directly generated from clinical swabs, while the remaining sequences were generated from viruses grown either in cells (n = 331) and eggs (n = 83). The virus source for HA sequence was unclear for the remaining 38 high-yield variants.

Among predicted as HY[both], 294 were CA/09-like viruses (38.6%), while 467 were WI/19-like viruses (61.4%). These high-yield strains were not geographically clustered and were scattered across the phylogenetic trees, without clear association with any particular HA lineages (Fig. 3B). However, the number of HY[both] strains increased significantly after the emergence of WI/19-like variants (Fig. 3D). Specifically, 256 out of 2,198 (11.65%) viruses in 2019 and 386 out of 895 (43.13%) viruses in 2020 were estimated to be HY[both] strains. MAIVeSS analysis predicted that the vaccine strain WI/19 has an increase of approximately 105-fold and 23-fold in virus yield in cell and eggs, respectively compared to CA/04.

**Table 1 | Antigenicity associated residues selected by MAIVeSS for the A(H1N1)pdm09 viruses**

| Residue[a] | Bootstrap[b] | $w^{global\ c}$ (±SD) | Residue | Bootstrap | $w^{global}$ (±SD) | Residue | Bootstrap | $w^{global}$ (±SD) |
|---|---|---|---|---|---|---|---|---|
| 45 | 100 | 0.1019 (0.0051) | 133a | 89 | 0.0419 (0.0034) | 194 (Sb, RBS)[d] | 100 | 0.0918 (0.0017) |
| 46 | 93 | 0.0337 (0.0018) | 142 (Ca2) | 100 | 0.1033 (0.0026) | 198 (Sb) | 100 | 0.0631 (0.0007) |
| 53 | 100 | 0.3728 (0.0018) | 144 (Ca2) | 100 | 0.1322 (0.0037) | 208 (Ca1) | 100 | 0.0689 (0.0037) |
| 60 | 100 | 0.4071 (0.0118) | 158 (Sa, Gly) | 100 | 0.2550 (0.0010) | 212 | 100 | 0.2921 (0.0021) |
| 63 (Gly) | 100 | 0.2220 (0.0067) | 159 (Sa) | 100 | 0.3895 (0.0131) | 214 | 88 | 0.0262 (0.0013) |
| 65 | 100 | 0.0748 (0.0035) | 160 (Sa) | 94 | 0.0312 (0.0014) | 219 | 94 | 0.0530 (0.0037) |
| 96 | 100 | 0.0424 (0.0015) | 165 (Sa) | 100 | −0.0286 (0.0016) | 271 | 100 | −0.0244 (0.0035) |
| 125c | 100 | 0.2380 (0.0024) | 188 (Sb) | 90 | 0.1739 (0.0105) | 273 | 100 | 0.0684 (0.0021) |
| 129 (Gly) | 100 | −0.1397 (0.0057) | 189 (Sb) | 100 | 0.1138 (0.0036) | 274 | 100 | 0.6435 (0.0050) |
| 133 | 89 | 0.5540 (0.0325) | 193 (Sb, RBS) | 100 | 0.1502 (0.0017) | 290 | 82 | 0.0649 (0.0036) |

The location of these residues in the HA protein structure are illustrated in Fig. 2B.

*SD*, standard deviation.

[a]*ABS* antibody binding sites, *RBS* receptor binding site, *Gly* N-linked glycosylation.

[b]The bootstrap values were derived from 100 independent experiments, each with 80% of the training data.

[c]Global weight ($w^{global}$) learned from the MTL-GGSL model in the MAIVeSS, and the absolute local weight for each individual task ($w^{local}$) is available from Supplementary Table S5.

[d]The position in which a potential egg adapted amino acid substitution L194I was reported to affect antigenic properties[60].

Multiple amino acid substitutions associated with high yield were observed in the HYboth strains, but the specific amino acid substitutions varied across influenza seasons and even within the same season, depending on the strain (Supplementary Data 13). However, after the 2018–2019 influenza season, viruses with HA K133aN, N159K/D/S, K166Q, S206T, and/or K214R were more likely to be high-yield strains (Fig. 4A).

To validate the predictions of our model, we synthesized the HA and NA genes of 4 viruses predicted as having desirable features, and subsequently generated 4 reassortant viruses (i.e. rgSP/16, rgCQ/17, rgBRU/19 and rgMAS/20) with PR8 as the backbone and determined their antigenic and yield phenotypes. Antigenically, 2 reassortant viruses were experimentally confirmed to be CA/04-like and the other 2 WI/19-like (Fig. 4B). All 4 reassortant viruses grew to final titers of >10$^8$ TCID$_{50}$/mL in both eggs and cells, which were at least 100-fold higher than the WT virus, which is a reassortant with wild-type HA and NA genes from CA/04 and six internal genes from PR8. (Fig. 4C). These results corroborated both antigenicity and yield predicted by the MAIVeSS model.

Taken together, our findings indicate that the high-yield trait of A(H1N1)pdm09 viruses was distributed across different genetic clusters and has become more prevalent since 2018. Our experimental results confirm MAIVeSS's ability to identify antigenic matches and high-yield vaccine strains for A(H1N1)pdm09 viruses.

**Diversifying influenza virus glycan binding profile facilitates the acquisition of high-yield properties**

It is well-documented that CA/04 exhibits an exclusive binding preference for 6'SLN and does not bind to 3'SLN[23]. Here we hypothesized that a small portion of naturally circulating A(H1N1)pdm09, such as those we predicted as CVVs, acquired high-yield properties by binding to additional sialylated glycan receptors, particularly 3'SLN, or by increasing their glycan binding avidity to 6'SLN. To test this, we conducted BLI for 6A(H1N1)pdm09 variants, including LY$^{both}$ MI/15 and HY$^{both}$ WI/19, as well as 4 HY$^{both}$ vaccine candidates predicted by MAIVeSS. The results showed that 3 of the MAIVeSS predicted CVVs, rgSP/16, rgCQ/17 and rgMAS/20, bound to both 3'SLN and 6'SLN, whereas MI/15, WI/19, and one predicted CVV, rgBRU/19, bound only to 6'SLN (Fig. 5A).

Of the residues linked to the high-yield traits of WI/19 and the four MAIVeSS selected viruses, only about half were conserved (Fig. 4A). However, HA N159K, K166Q, and S206T were consistently present in most of the naturally occurring high-yield strains (Supplementary Data 12) and high-yield mutants from our mutagenesis study (Supplementary Data 4).

We further investigated the effect of HA N159K, K166Q, and S206T on glycan binding affinity by conducting structural modeling based on the crystal structure of CA/04 HA complexed with 6'SLN and 3'SLN (Fig. 5B). In both complex structures, N159 was substituted with K, and energy minimization was performed to detect any possible allosteric structural changes that could affect ligand binding. In the HA:3'SLN structure, the sidechain of K159 flips toward the 190-helix, forming hydrogen bonds with the sidechains of both Q196 and Q192. This could cause a shift or tilt in the orientation of the 190-helix, resulting in a more compact receptor binding pocket and stronger binding with 3' SLN. In contrast, 6'SLN in the HA:6'SLN complex closely interacts with the 190-helix even in the original CA/04 structure, so the K159 mutation does not significantly enhance the binding of 6'SLN binding to HA. Additionally, K166Q may impact the conformation of the 130-loop, while S206T substitution has the potential to modify the structural conformation of the 220-loop (Supplementary Fig. 7), thus affecting the binding of HA to glycan receptors. Therefore, our modeling analysis supported that these three substitutions in HA can substantially increase the binding affinity of 3'SLN without major impact on the binding of 6'SLN.

In summary, diversity at the HA RBS of A(H1N1)pdm09 viruses can enhance virus yields in both cell and egg substrates by increasing sialylated glycan binding avidity or diversifying virus binding to different sialylated glycan receptors.

## Discussion

In this study, we developed MAIVeSS, a machine learning based framework, that can accurately predict both antigenicity and growth phenotypes based on HA protein sequences. The training dataset consisted of a library of 189 mutant viruses generated by epPCR-based reverse genetics targeting residues 126–244 (H3 numbering). We observed that acquisition of HA N159K, a key marker for antigenic drift according to our model, led to changes in antigenicity from CA/09 to WI/19, as determined by post-infection ferret antisera, consistent with published reports[21,22] and facilitated acquisition of the high-yield trait in a significant proportion of A(H1N1)pdm09 epidemic strains during recent influenza seasons (Fig. 3E). While our current model focuses on HA, it is important to note that antigenic drift of neuraminidase (NA) has also been well-documented in A(H1N1) and A(H3N2) influenza viruses[24,25]. In addition, it's worth noting that the antigenicity data used in our model training were derived from ferret antisera generated from native ferrets. While the antigenicity data reflected by sera generated from native ferrets and human sera without virus priming (such as infants) are generally comparable, it's important to note that they can differ from those generated from individuals with pre-existing

**Table 2 | Amino acid substitutions associated with yield traits selected by MAIVeSS for the A(H1N1)pdm09 viruses**

| Residue[a,b,c] | Yield trait[g] | | | | | | | | | | | |
|---|---|---|---|---|---|---|---|---|---|---|---|---|
| | HY[cell] | Bootstrap[f] | Weight (±SD) | HY[pdm] | Bootstrap | Weight (±SD) | LY[cell] | Bootstrap | Weight (±SD) | LY[pdm] | Bootstrap | Weight (±SD) |
| **126** | P to SP | 81 | 0.5705 (0.2311) | P to SP | 80 | 1.6838 (0.6509) | – | | | – | | |
| 131[d] | –[e] | | | P to P | 100 | 0.2478 (0.2018) | P to P | 99 | −0.2527 (0.2014) | – | | |
| 132 | P to NP | 87 | 0.4785 (0.2082) | P to NP/SP | 85 | 0.7507 (0.3073) | P to SP | 99 | −0.5006 (0.1974) | – | | |
| **133a** | P to NP/P | 99 | 0.5869 (0.0892) | P to NP/P | 100 | 1.7985 (0.4504) | – | | | | | |
| **137** | | | | | | | SP to NP/P | 94 | −1.4894 (0.6504) | SP to NP/P | 95 | −0.4997 (0.1827) |
| **138** | SP to P | 82 | 0.0693 (0.1157) | SP to P | 82 | 0.9026 (0.3905) | SP to P | 95 | −0.3991 (0.2088) | | | |
| 140 (Ca2) | SP to P | 100 | 0.1603 (0.3925) | | | | SP to NP | 98 | −0.2150 (0.7936) | | | |
| **141 (Ca2)** | P to P | 100 | 0.9139 (0.4540) | P to NP/P | 100 | 1.0663 (0.3840) | P to NP | 100 | −0.8497 (0.2333) | | | |
| **142 (Ca2)** | SP to NP/SP | 93 | 0.2102 (0.4026) | SP to NP | 96 | 1.1147 (0.3030) | SP to P | 96 | −1.1819 (0.4838) | SP to P/SP | 96 | −0.1094 (0.2109) |
| **144 (Ca2)** | SP to NP/SP/P | 81 | 0.3179 (0.1226) | SP to NP/SP/P | 82 | 1.1002 (0.4648) | – | | | | | |
| 146 | – | | | P to NP/SP | 99 | 0.4318 (0.1459) | P to P/NP/SP | 100 | −0.5463 (0.3416) | P to P | 84 | −0.0683 (0.0289) |
| 149 | – | | | P to NP/SP | 96 | 0.3256 (0.3971) | P to NP/SP | 99 | −0.0439 (0.1606) | | | |
| **157 (Sa)** | P to NP/P | 100 | 0.5483 (0.6302) | P to NP/P | 100 | 1.1876 (0.3793) | – | | | P to NP | 88 | −1.2852 (0.6344) |
| **159 (Sa)** | P to P | 99 | 0.7856 (1.1559) | P to P | 97 | 0.0379 (0.1499) | P to NP | 85 | −0.7152 (0.3527) | SP to P | 100 | −0.1045 (0.5025) |
| **162 (Sa)** | – | | | | | | SP to P | 99 | −0.0396 (0.3050) | | | |
| **166 (Sa)** | P to NP/P | 100 | 0.4145 (0.8401) | P to NP/P | 99 | 1.5899 (1.2154) | – | | | | | |
| 167 (Sa) | – | | | P to P | 100 | 0.8343 (0.2141) | P to P | 100 | −0.1047 (0.353) | | | |
| **169 (Ca1)** | NP to NP | 100 | 0.5871 (0.4362) | NP to NP | 100 | 1.8680 (0.5976) | NP to P | 100 | −0.3385 (0.2147) | NP to P | 99 | −0.9964 (0.1964) |
| 173 (Ca1) | – | | | | | | SP to P | 100 | −0.5564 (0.2064) | SP to P | 100 | −0.9965 (0.3552) |
| 174 | – | | | | | | P to P | 99 | −1.1321 (0.2201) | P to P | 99 | −0.1380 (0.6943) |
| 178 | – | | | NP to NP | 99 | 0.1548 (0.0990) | NP to NP | 85 | −0.1794 (0.2453) | | | |
| **182** | NP to NP | 85 | 1.2553 (0.5159) | NP to NP | 84 | 2.4218 (1.0087) | – | | | | | |
| **188 (Sb)** | P to SP/P | 83 | 0.8182 (0.3406) | P to SP | 89 | 0.1816 (0.0669) | P to NP | 83 | −0.2922 (0.2709) | P to NP/P | 88 | −1.2478 (0.5429) |
| 189 (Sb) | SP to NP | 100 | 0.2487 (0.4659) | SP to P | 100 | 0.7942 (0.1624) | SP to P | 99 | −0.2301 (0.1982) | SP to NP | 100 | −0.0306 (0.0891) |
| 192 (Sb) | – | | | P to NP | 98 | 0.3159 (0.2751) | P to NP | 93 | −1.1719 (0.4636) | | | |
| **193 (Sb, RBS)[d]** | P to P | 88 | 1.8718 (0.9340) | P to P/NP | 82 | 2.5413 (1.1365) | P to SP/NP | 82 | −0.0369 (0.6627) | P to SP | 86 | −0.3778 (0.1401) |
| **197 (Sb)** | – | | | | | | P to P | 97 | −0.6787 (0.3430) | P to P | 93 | −0.2856 (0.4440) |
| **198 (Sb)** | – | | | SP to NP/P | 100 | 1.1147 (0.9888) | SP to SP/NP/P | 100 | −0.1013 (0.4861) | SP to SP | 81 | −0.2478 (0.2074) |
| **205** | SP to SP | 82 | 0.9085 (0.3800) | SP to SP | 80 | 1.2537 (0.3943) | SP to P | 98 | −0.2549 (0.7150) | SP to P | 100 | −0.5482 (0.4787) |
| **206 (Ca1)** | – | | | | | | P to P | 99 | −0.5541 (0.2891) | P to P | 100 | −0.6986 (0.7722) |
| **211** | | | | | | | P to P | 100 | −0.4308 (0.1808) | P to P | 99 | −0.6789 (0.3159) |
| **212** | | | | | | | P to NP/P | 97 | −0.8865 (0.1492) | P to NP/P | 100 | −0.7254 (0.2640) |
| **214** | P to NP/P | 100 | 0.3179 (0.1253) | P to NP/P | 100 | 1.5003 (0.5523) | – | | | NP to NP | 98 | −0.0765 (0.1014) |
| 219 | – | | | NP to P | 95 | 0.2334 (1.0338) | NP to NP/P | 100 | −0.6312 (0.2283) | P to P | 99 | −0.6276 (0.3037) |
| **222 (RBS)** | – | | | | | | P to P | 100 | −0.1763 (0.1970) | P to SP | 99 | −0.5636 (0.2010) |
| 225 (Ca2, RBS)[d] | P to SP | 100 | 0.2013 (0.2212) | P to P | 96 | 0.8162 (0.3808) | – | | | P to P | 93 | −0.0520 (0.2880) |
| 229 | P to P | 97 | 0.0023 (0.1321) | – | | | – | | | | | |
| **237** | – | | | | | | NP to NP | 100 | −0.1318 (0.2421) | NP to NP | 100 | −0.6804 (0.2477) |

The location of these residues in the HA protein structure are illustrated in Fig. 2C.
[a] ABS antibody binding site, RBS receptor binding site.
[b] Each amino acid is assigned to one of the three groups: nonpolar (NP) (V, L, I, M, C, F, W, and Y), small polar (SP) (G, A, and P), and polar/charged (P) (S, T, N, Q, H, D, E, K, and R) based on their biophysical properties.
[c] The substitutions associated with HY[pdm] or LY[pdm] were in bold.
[d] The positions reported to potentially harbor an egg adapted amino acid substitution, of which D131E and D225G/N were reported to affect antigenic properties[60].
[e] "—" denotes it is unknown how this residue will affect yield.
[f] The bootstrap values were derived from 100 independent experiments, each with 80% of the training data.
[g] The amino acid substitutions highlighted in bold are those anticipated to be associated with each specific yield phenotype.

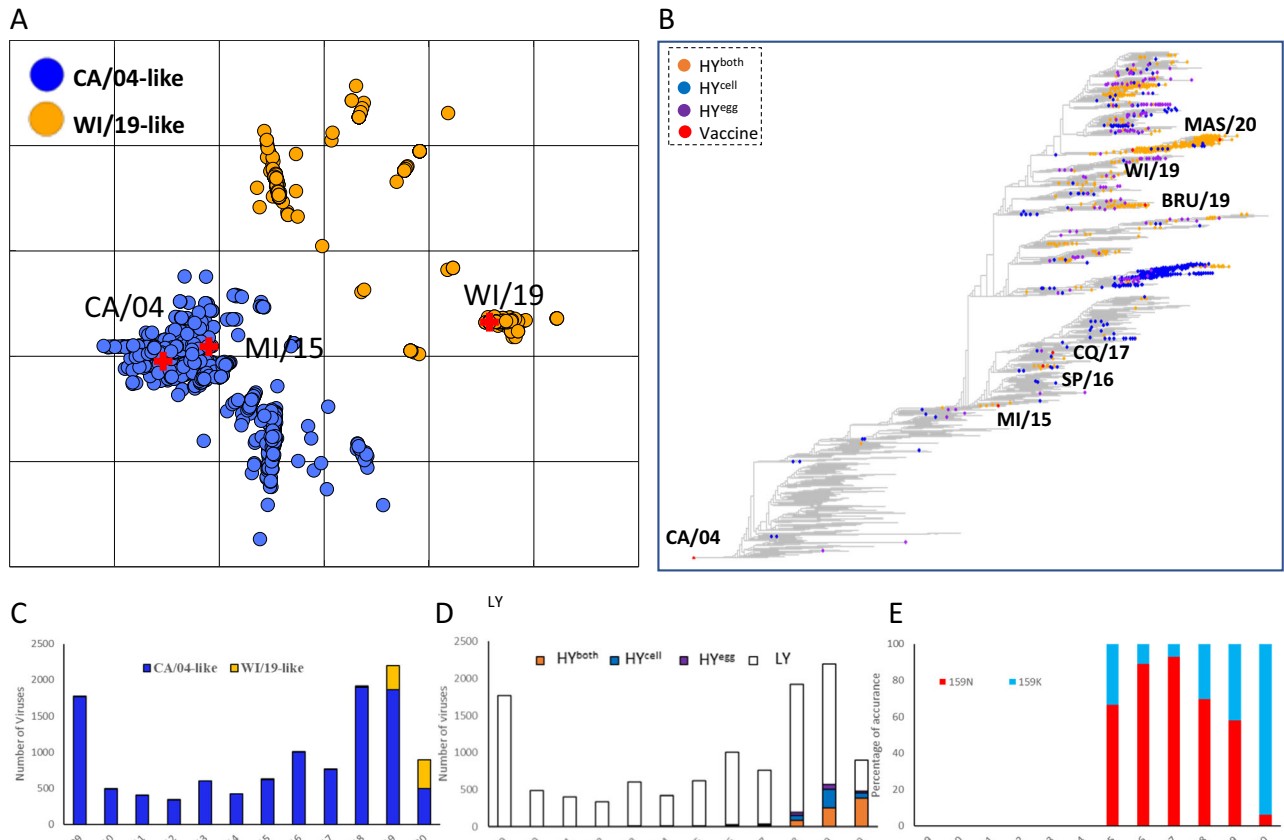

**Fig. 3 | Antigenicity and yield of A(H1N1)pdm09 viruses from 2009 to 2020 as predicted by MAIVeSS. A** Two major antigenic variant clusters for A(H1N1)pdm09 viruses were visualized by antigenic cartography. **B** Visualization of the distribution of high-yield viruses within the phylogenetic tree of A(H1N1)pdm09 viruses. Vaccine strains, high-yield viruses in both cell and egg (HY^both), high-yield viruses in only cell (HY^cell), and high-yield viruses in only egg (HY^egg) are indicated by red, orange, blue, and purple, respectively, in the phylogenetic tree of A(H1N1)pdm09 viruses. **C** The number of two antigenically distinct variants, CA/04-like and WI/09-like, across the 2009 to 2020 influenza seasons. **D** The proportion of HY^both, HY^cell, HY^egg, and low yield viruses in both cell and egg (LY) across the 2009 to 2020 influenza

seasons. **E** The presence of N159K substitutions in the HA of pdmH1N1 high yield viruses from 2009 to 2020. Shown are the number of HA sequences analyzed by the percentage of the indicated mutations. CA/04 A/California/04/2009(H1N1), MI/15 A/Michigan/45/2015(H1N1), SP/16 A/Saint-Petersburg/RII57/2016, CQ/17 A/Chongqing-Yuzhong/SWL1453/ 2017(H1N1), BRU/19 A/Brunei/25/2019(H1N1), WI/19 A/Wisconsin/588/2019(H1N1), MAS/20 A/Malaysia/33075487/2020. CA/04-like denotes A/California/04/2009(H1N1)-like A(H1N1)pdm09 viruses whereas WI/19-like does A/Wisconsin/588/2019(H1N1)-like A(H1N1)pdm09 viruses. Source data are provided as a Source Data file.

immunity, particularly in adults. This is because adults commonly have a complicated influenza priming history with multiple infections and/ or vaccinations, which can significantly affect their immune response[26]. As such, our ongoing efforts are aimed at expanding the MAIVeSS prediction capacity to include both HA and NA proteins, as well as human serological data for training antigenicity analyses.

To determine if the high-yield trait correlates with glycan substructure binding properties, we analyzed the receptor-binding properties of all 189 mutant viruses. The glycan profiling analysis conducted on 43 high-yield mutants suggested that diversifying glycan binding profiles could enhance virus replication in both eggs and cells. Specifically, increased binding avidities to SA2-6Gal results in higher virus yield in mammalian cells, while broadening glycan binding capabilities to SA2-3Gal or sLe^x improves virus yield in eggs (Supplementary Data 4). Our studies indicate that a small subset of A(H1N1)pdm09 epidemic viruses naturally bind to both SA2-6Gal and SA2-3Gal, allowing them to replicate efficiently in both cells and eggs without adaptation. On the other hand, similar to CA/04 and MI/15, some high-yield strains (e.g. WI/19) bind only to 6′SLN but not 3′SLN (Fig. 5A), indicating that other glycan substructures present in eggs and/or cells may be involved in the high-yield trait for these epidemic viruses. It should be noted that both virus and host factors, such as innate

immune responses and virus fitness, in addition to virus-receptor binding can impact virus replication in mammalian cells. Further studies are needed to investigate these possibilities.

Both SA2-6Gal and SA2-3Gal receptors are expressed in MDCK cells and chicken embryonated eggs. However, SA2-3Gal receptors are predominantly expressed in eggs while MDCK cells contain a similar amount of SA2-6Gal and SA2-3Gal[27]. In addition to SA2-3Gal and SA2-6Gal, neutral glycans such as high-mannose glycans and glycans terminated with Gal and GalcNAc are also commonly found in eggs. Mass spectrometry analyses showed some glycans in eggs are fucosylated[28]. CA/04, the prototype A(H1N1)pdm09 virus which showed poor replication in both MDCK cells and eggs, had a strong binding preference for SA2-6Gal and did not bind to SA2-3Gal[29]. In humans, there is no direct selection pressure to increase cell-based or egg-based replication efficiency. Thus, our findings suggested that ad hoc substitutions at the HA RBS across A(H1N1)pdm09 viruses likely enabled a subset of these variants to expand their binding preference from SA2-6Gal to both SA2-6Gal and SA2-3Gal, resulting in the acquisition of a high-yield trait. This study illustrates the feasibility of selecting HA sequences from naturally circulating strains as high yield candidates for recombinant vaccine development, by eliminating the need for further engineering.

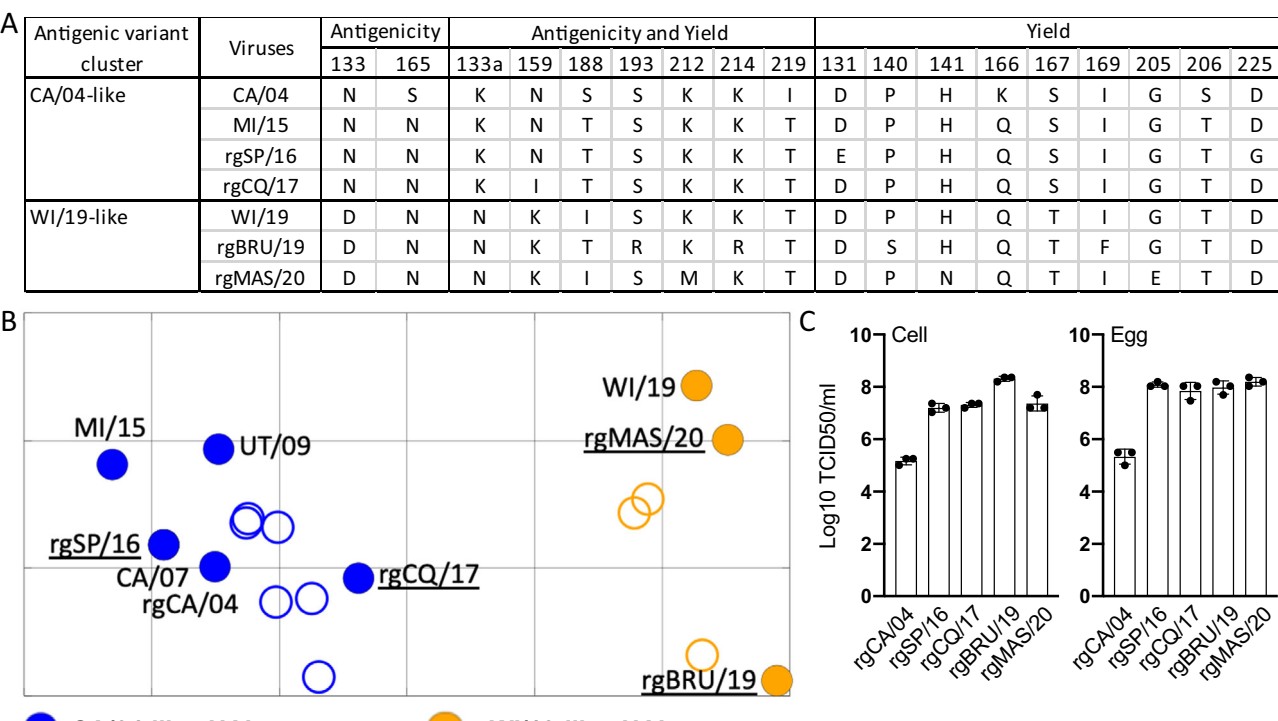

| Antigenic variant cluster | Viruses | Antigenicity | | Antigenicity and Yield | | | | | | | Yield | | | | | | | | |
|---|---|---|---|---|---|---|---|---|---|---|---|---|---|---|---|---|---|---|---|
| | | 133 | 165 | 133a | 159 | 188 | 193 | 212 | 214 | 219 | 131 | 140 | 141 | 166 | 167 | 169 | 205 | 206 | 225 |
| CA/04-like | CA/04 | N | S | K | N | S | S | K | K | I | D | P | H | K | S | I | G | S | D |
| | MI/15 | N | N | K | N | T | S | K | K | T | D | P | H | Q | S | I | G | T | D |
| | rgSP/16 | N | N | K | N | T | S | K | K | T | E | P | H | Q | S | I | G | T | G |
| | rgCQ/17 | N | N | K | I | T | S | K | K | T | D | P | H | Q | S | I | G | T | D |
| WI/19-like | WI/19 | D | N | N | K | I | S | K | K | T | D | P | H | Q | T | I | G | T | D |
| | rgBRU/19 | D | N | N | K | T | R | K | R | T | D | S | H | Q | T | F | G | T | D |
| | rgMAS/20 | D | N | N | K | I | S | M | K | T | D | P | N | Q | T | I | E | T | D |

**Fig. 4 | Validation of the predicted antigenic and yield properties for the MAIVeSS-selected vaccine viruses. A** Amino acids located in the residues associated with antigenicity and yield properties for three A(H1N1)pdm09 vaccines (i.e., CA/04, MI/15, and WI/19) selected by WHO and the four vaccine candidates selected by MAIVeSS. **B** Antigenic cartography of A(H1N1)pdm09 viruses and the four vaccine candidates selected by MAIVeSS. The position of filled circles represents the antigenic properties derived by the HAI data with ferret antisera (Table 3), and those in open circles represent the predictive antigenic properties by MAIVeSS. The vaccine strains were marked in gold, and the vaccine candidates and other epidemic A(H1N1)pdm09 viruses were marked in blue. **C** Replication efficiency of the four vaccine candidates selected by MAIVeSS. *N* = 3 biologically independent samples were used in the experiments, and data are presented as mean values ± standard deviation (SD). rgCA/04 A/California/04/2009(H1N1)(HA,NA)xPR8, CA/07

A/California/07/2009(H1N1), UT/09 A/Utah/20/2009(H1N1), MI/15 A/Michigan/45/2015(H1N1), WI/19 A/Wisconsin/588/2019(H1N1), rgSP/16 A/Saint-Petersburg/RII57/2016, rgCQ/17 A/Chongqing-Yuzhong/SWL1453/2017, rgBRU/19 A/Brunei/25/2019 (H1N1)(HA,NA)xPR8, rgMAS/20 A/Malaysia/33075487/2020 (H1N1)(HA,NA)xPR8. CA/04-like denotes A/California/04/2009(H1N1)-like A(H1N1)pdm09 viruses whereas WI/19-like does A/Wisconsin/588/2019(H1N1)-like A(H1N1)pdm09 viruses. Antigenic cartography was constructed by using AntigenMap (http://sysbio.missouri.edu/AntigenMap)[61], which employed a low-reactor cutoff of 1:10. To minimize noise in the HAI data and reflect antigenic distances embedded in the data, low-rank matrix completion and multiple dimensional scaling were utilized to generate the map. Each unit in the antigenic map corresponded to a log2 unit In the HAI titers. Source data are provided as a Source Data file.

In summary, the data from the proof-of-concept experiments in this study confirmed that MAIVeSS enables rapid selection of antigenically matched and high-yielding influenza strains directly from clinical isolates as potential seed viruses to accelerate vaccine production and facilitate timely supply of seasonal vaccines.

## Methods

### Ethics statement

Animal study protocols were reviewed and approved by the Institutional Animal Care and Use Committee at Mississippi State University (#14-039) and University of Missouri-Columbia (#9656). All animal experiments were performed in an animal biosafety level 2 (ABSL2) research facility at Mississippi State University. Standard operating procedures for work with infectious influenza viruses were approved by the Institutional Biosafety Committee at Mississippi State University (#16-09 and #022-16) and University of Missouri-Columbia (#19-09) and performed under BSL2 conditions.

### Machine-learning assisted influenza vaccine strain selection framework (MAIVeSS)

Machine learning models have been shown to be effective in identifying antigenicity associated features in protein sequences from

different subtypes of influenza A viruses[30–35], We developed machine learning models to identify the specific sequence features in HA proteins that determine three important phenotypes: antigenicity, yield in cells and eggs, and receptor binding. To achieve this, we trained our models on large datasets of HA protein sequences and associated phenotype information. We also developed a quantitative function that allows us to measure the distances between sequences based on their phenotypic characteristics. Our ultimate goals for these machine learning models are to identify: 1) mutations in the HA RBS that affect virus antigenicity; 2) mutations in the HA RBS that increase or decrease virus yields in cells and/or eggs; and 3) specific glycan substructures (glycan motifs) on the surface of cells or eggs that are associated with increased yields of influenza virus. By achieving these goals, we hoped to gain a better understanding of the molecular determinants of these important viral phenotypes and to identify potential targets for the development of improved influenza vaccines.

We approached the problem of identifying genetic features associated with influenza virus phenotypes using a sparse learning model. Mathematically, this model involves a linear regression loss function with regularization, which allows us to determine the most relevant genetic features associated with a given phenotype. The sparse learning model combines a least squares loss with a regularized

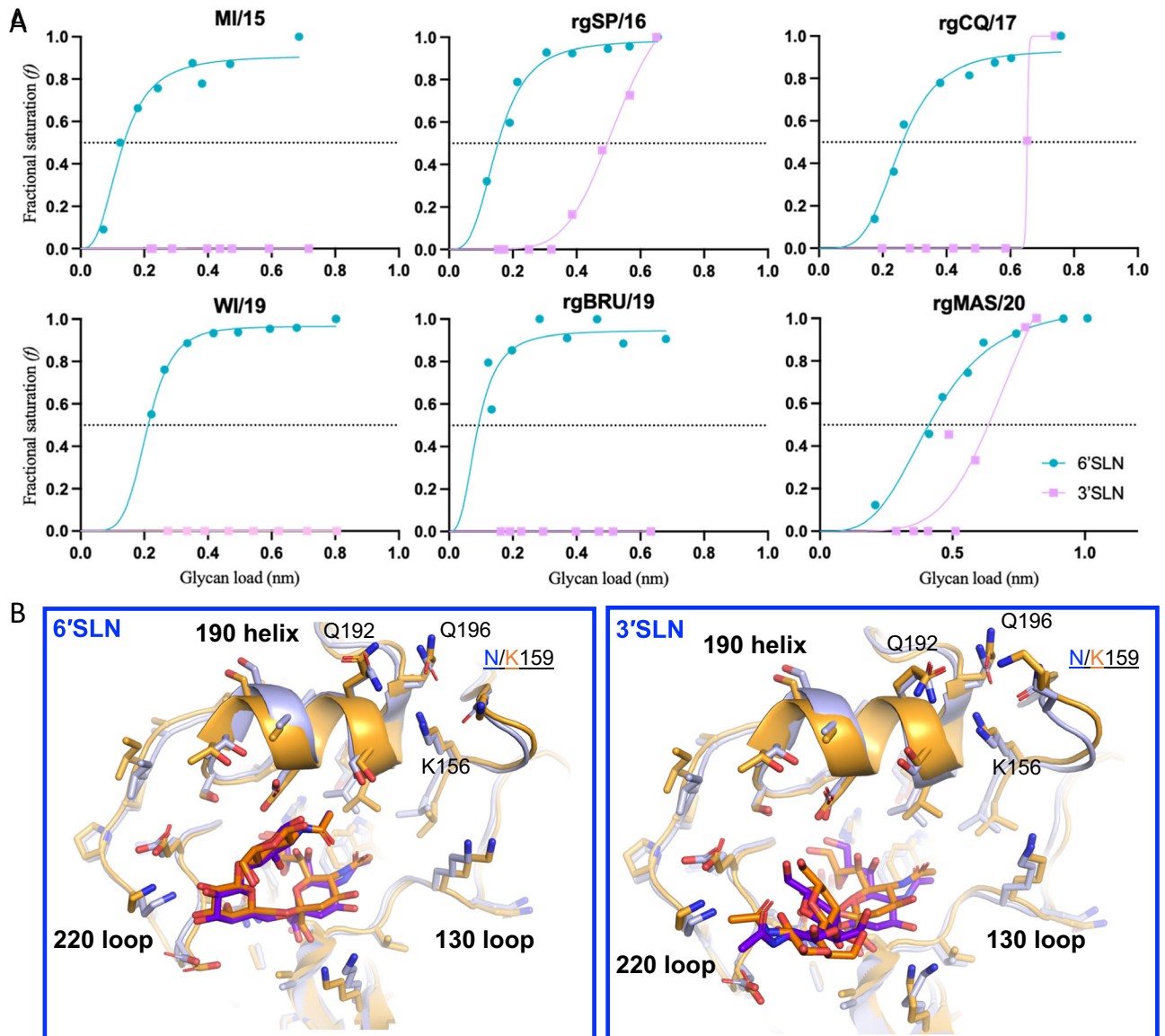

**Fig. 5 | Glycan binding properties for the MAIVeSS-selected vaccine viruses.**
**A** The binding avidity to the synthetic glycan analogs, 3'SLN and 6'SLN for two A(H1N1)pdm09 vaccines (i.e., MI/15, and WI/19) selected by WHO and the four vaccine candidates selected by MAIVeSS. The virus abbreviations are shown in the legend of Fig. 4. The virus receptor binding affinities were determined by BLI with an Octet RED instrument (Pall ForteBio, Menlo Park, CA). Two biotinylated glycan analogs, Neu5Acα2-3Galβ1-4GlcNacβ-PAA-biotin (3'SLN) and Neu5Acα2-6Galβ1-4GlcNacβ-biotin (6'SLN) (Lectinity Holdings, Moscow, Russia) were used. The glycans ranging from 0.05 to 0.5 ug/ml were preloaded onto streptavidin-coated biosensors at up to 0.3 μg/ml for 5 min in 1 × kinetic buffer (Pall FortéBio, Menlo Park, CA). Each test virus was diluted to a final concentration of 100 pM. Responses were normalized by the highest value obtained during the experiment, and binding curves were fitted by using the binding-saturation method in GraphPad Prism 8 (https://www.graphpad.com/scientific-software/prism/). The normalized response curves report the fractional saturation ($f$) of the sensor surface as described in elsewhere[58]. The horizontal dashed line represents half of the fractional saturation ($f = 0.5$). Results showed that rgSP/16, rgCQ/17, and rgMAS/20 bound to both 3'SLN

and 6'SLN, whereas MI/15, WI/19, and rgBRU/19 bound only to 6'SLN. The rgBRU/19 showed an increased binding avidity to 6'SLN (RSL$_{0.5}$ = 0.085) compared to MI/15 (RSL$_{0.5}$ = 0.135). The rgSP/16, rgCQ/17, and rgMAS/20 still exhibited a stronger binding affinity to 6'SLN (RSL$_{0.5}$ = 0.147, 0.263, and 0.134, respectively) compared to 3'SLN (RSL$_{0.5}$ = 0.498, 0.653, and 0.347). **B** Structural modeling suggesting the amino acid substitution N159K facilitates the binding affinity of HA to SA2-3Gal. The three-dimensional structure of HA protein was modeled based on the crystal structure of CA/04 HA in complex with 6'SLN (PDB ID# 3UBN) and 3'SLN (PDB ID# 3UBQ). Coot was used to introduce the desired mutation to the three subunits of a HA trimer. The mutated coordinates were subsequently refined by energy minimization using Phenix. Structure figures were made using Pymol (The PyMOL Molecular Graphics System, Version 1.3, Schrödinger, LLC). MI/15 A/Michigan/45/2015(H1N1), WI/19 A/Wisconsin/588/2019(H1N1), rgSP/16 A/Saint-Petersburg/RII57/2016, rgCQ/17 A/Chongqing-Yuzhong/SWL1453/2017, rgBRU/19 A/Brunei/25/2019 (H1N1)(HA,NA)xPR8, rgMAS/20 A/Malaysia/33075487/2020 (H1N1)(HA,NA)xPR8, 3'SLN Neu5Acα2-3Galβ1-4GlcNacβ-PAA-biotin, 6'SLN Neu5Acα2-6Galβ1-4GlcNacβ-PAA-biotin. Source data are provided as a Source Data file.

---

term and takes into account genetic distance matrices among HA proteins or glycan sequences (denoted as X), phenotypic differences (denoted as y), and sample numbers (denoted as N). This approach enables us to identify the key genetic features that contribute to different influenza virus phenotypes, such as antigenicity, yield, and receptor-binding.

The objective of our sparse learning model is to solve:

$$\min L(X,y,w) + \lambda R(w), \qquad (1)$$

where $L(X,y,w)$ is the loss function, $\lambda$ is a pre-defined regularization parameter, $R(w)$ denotes the regularization term, and $w$ denotes the

numerical weights of individual features (either a single residue or a group of neighboring residues). Absolute values of the weights indicate the impact of each mutation of a specific feature to phenotypes (i.e., antigenic, yield, and receptor-binding properties). The larger the absolute weight, the greater the impact.

Based on the features learned from sparse learning, we developed a predictive model to assess antigenic or yield properties given HA sequences. Specifically,

$$\hat{y} = xw, \tag{2}$$

where $\hat{y}$ is the predicted phenotypic distance (either antigenicity or yield) between the two viruses; x is the feature distance vector; and w is the weight vector for those features, which can be associated with either antigenicity or yield.

## Multi-task learning group-guided sparse learning (MTL-GGSL) model

To address the challenges associated with integrating serological data generated from different platforms (e.g., turkey and guinea red blood cells), we utilized a Multi-Task Learning (MTL) approach with Group Graphical Sparse Learning (GGSL) to analyze antigenicity. This approach allowed us to consider both N-linked glycosylation and amino acid features when analyzing the data. MTL allows us to learn multiple related tasks (i.e., analyzing antigenicity from different serological platforms) simultaneously, while GGSL considers the dependencies between different groups of features to improve the accuracy of the analysis. By utilizing MTL-GGSL, we were able to overcome the challenges associated with integrating data from different platforms and provide a more comprehensive analysis of antigenicity[20,36].

One advantage of using the group LASSO regularization in MTL-GGSL for antigenicity analyses is that it encourages multiple predictors from related tasks to share a subset of features. This is in contrast to the LASSO regularization, which may lead to sparse solutions where only a few features are selected for each task independently. Our previous study has shown that incorporating information on N-linked glycosylation can improve the performance of sparse learning models in predicting antigenic properties of influenza viruses[20]. By adopting MTL-GGSL, we are able to integrate information on both glycosylation and amino acid sequences from serological data generated using different platforms, which can further enhance the accuracy of our predictive models for influenza antigenicity.

Specifically, we define

$$L(X, y, w) = \frac{1}{2} ||Y - XW||_F^2, \tag{3}$$

and

$$\lambda R(W) = \lambda_1 R_1(W) + \lambda_2 R_2(W) + \lambda_3 R_3(W), \tag{4}$$

and the model is formulated as:

$$\min_{W} \frac{1}{2} ||Y - XW||_F^2 + \lambda_1 \sum_{j=1}^{p} ||W_{j \cdot}||_1 + \lambda_2 \sum_{t=1}^{k} \sum_{l=1}^{q} \alpha_l ||W_{G_l, t}||_2$$
$$+ \lambda_3 \sum_{t=1}^{k} \sum_{l=1}^{q} \alpha_l ||W_{G_l, t}||_1, \tag{5}$$

where $\lambda_1$, $\lambda_2$, and $\lambda_3$ are regularization parameters, j is the subscript for feature, p is the total number of features, $G_l$ denotes feature group, q is the number of feature groups, $\alpha_l = \sqrt{m_l}$ is the weight of feature group $G_l$; $W_j$ denotes the weights for the $j$-th feature among different tasks, and $W_{G_l, t}$ as the weight for feature group $G_l$ of the $t$-th task. Alternating

Direction Method of Multipliers (ADMM)[37] was employed to solve the optimization problem.

## The generalized hierarchical sparse model (GHSM)

To consider synergistic effects of multiple features on the phenotypes, we adopt GHSM[38] in this study. The GHSM model aims to minimize: $L(W) + \sum_{k=1}^{K} \frac{\lambda}{\alpha^k} ||W^{(k)}||_1$. GHSM model solves the following objective:

$$\min_{W} \frac{1}{2} \left\| y - \sum_{k=1}^{K} \sum_{i_1, \cdots, i_k}^{d} w^{(k)}_{<i_1, \cdots, i_k>} z^{(k)}_{<i_1, \cdots, i_k>} \right\|_2^2 + \sum_{k=1}^{K} \frac{\lambda}{\alpha^k} ||w^{(k)}||_1,$$
$$s.t. |w_i^{(1)}| \geq ||e_i^{(2)} \odot w^{(2)}||_1 \geq \cdots \geq \left\| e_i^{(K)} \odot w^{(K)} \right\|_1, i \in \mathbb{N}_d, \tag{6}$$

where $\lambda$ and $\alpha$ are two regularization parameters controlling the sparsity and the decay in the coefficients for interactions of different orders, $z^{(k)}_{<i_1, \cdots, i_k>} = x_{i_1} \odot x_{i2} \odot \cdots \odot x_{ik}$ denotes a data vector for the $k$-th order interaction corresponding to $<i_1, \cdots, i_k>$, an interaction index $<i_1, \cdots, i_k>$, where $i_1 < \cdots < i_k$, is an index to uniquely indicate the interaction among the covariates $i_1, \cdots, i_k$, $W$ denotes the set of parameters $\{w^{(k)}\}_{k=1}^{K}, w^{(k)} \in \mathbb{R}^{\binom{d}{k}}$ for $k = 1, \cdots, K$ is a vector of length $\binom{d}{k} = \frac{d!}{k!(d-k)!}$ with $w^{(k)}_{<i_1, \cdots, i_k>}$ as its element corresponding to the index $<i_1, \cdots, i_k>$, $||\cdot||_2$ denotes $l_2$ norm of a vector and $\odot$ denotes the element wise product of two vectors. The constrains associated with each covariate $i$ have a chain of $(K - 1)$ inequality constraints, and there is a total of $d$ chains. The application of these models for antigenicity, yield, receptor-binding are detailed as below.

## Antigenicity analyses

In this study, we used eight individual tasks, each corresponding to an individual HAI dataset, including those for seasonal A(H1N1) viruses (1977–2009), A(H1N1)pdm09 viruses (2009–2020), swine A(H1N1) viruses, and mutants generated from this study. In each task, the low-rank matrix completion algorithm was used to minimize data noise and the challenges derived from low reactors and missing values in the HAI datasets, and antigenic cartography was then used for antigenic distance calculation[20,39]. Two groups of features (i.e., amino acid mutations and N-glycosylation sites) were used in the model to quantify influenza virus antigenic distances. We defined 327 residue features and 6 N-glycosylation site features. GETAREA software (http://curie.utmb.edu/getarea.html) was used to predict whether these residues were on HA's surface. The A(H1N1)pdm09 three-dimensional HA structure (Protein Data Bank [PDB] identifier [ID] 3LZG) was used as the template. A total of 138 residues were predicted to be located at the HA protein's surface (Supplementary Data 14). All amino acid residues, with a variant rate >10%, were considered as non-conserved sites and included in the machine learning model. Finally, a total of 86 residues with 4 N-glycosylation sites were used as features in the machine learning model.

## Yield analyses

In this study, we analyzed the yield of 189 mutants compared with the parent WT CA/04, which is 6:2 reassortant virus, in both cells and eggs. To analyze the data, we utilized two groups of features: amino acid substitutions and N-glycosylation sites. Furthermore, we employed the GHSM approach to identify synergistic amino acid substitutions associated with virus yield in eggs or cells. Specifically, to ensure the feasibility of our analyses, we constrained the highest order to 3. It's worth noting that there were 3468 second-order interactions and 12,337 third-order interactions in our GHSM analyses.

## Glycan binding analyses

In this study, we used a glycan microarray with 75 glycoforms (Supplementary Fig. 1), which were grouped based on their internal and

terminal substructures and linkers into a matrix of 27 glycan substructure features. We then utilized the Multi-Task Learning with Group Graphical Sparse Learning (MTL-GGSL) approach to determine the substructures associated with yield traits in cells and eggs. In the model, we employed three groups of features, including terminal substructures ($n = 16$), internal substructures ($n = 8$), and base substructures linked to the array ($n = 3$).

### Model comparison, parameter optimization, and bootstrapping analyses

In order to ensure the robustness of our analyses on antigenicity, yield, and glycan binding, we compared MAIVeSS with five other sparse learning models, including the L1-norm regularized method (LASSO)[40], the L2-norm regularized method (RIDGE)[41], the sparse group LASSO method (SGL)[42], the L1- and L2-norm regularized method[43], and the L1- and L∞-norm Composite Absolute Penalties method (iCAP)[44] (Supplementary Data 1–3). The latter two models incorporate both L1- and L2-norm regularization.

For antigenicity analyses, our comparison additionally included three primary machine learning models mentioned in the literature, along with three deep learning approaches. The conventional machine learning methods consist of Support Vector Machine (SVM)[45–47], Naïve Bayes[14,48–50], and Random Forest[35,51,52]. The deep learning methods include Gated Recurrent Unit (GRU), Long Short-Term Memory (LSTM), and Transformer. As the source codes for these models from the literature are not publicly accessible, we utilized the machine learning models available in MATLAB package (R2023a) for comparison.

To develop and evaluate our models, we allocated 90% of our data for training and validation, and the remaining 10% for testing. The testing dataset was excluded from parameter optimization to avoid potential overfitting. Within the combined training and validation dataset, we employed 10-fold cross-validation by segregating the data, 90% for training and the remaining 10% for validation, to fine-tune our parameters and evaluate the training performance (Supplementary Figs. 8–13). As the results, we set $\lambda_1$ equals 2, $\lambda_2$ equals 0.01, and $\lambda_3$ equals 0.01 as optimal parameter for MTL-GGSL model, and $\lambda$ equals 0.0001 and $\alpha$ equals 10 for GHSM model.

To evaluate the performance between models in MAIVeSS and the previously mentioned comparison models, we assessed their RMSE and Pearson correlation coefficient between predicted values and ground truth for both training (using 10-fold cross-validation) and testing datasets for antigenicity, yield, and glycan binding analyses. For antigenicity analyses, we also recast it as an antigenic distance classification problem to assess the model's efficacy in identifying antigenic variants. A virus pair is classified as an antigenic variant if its paired antigenic distance is 4-fold or greater; otherwise, it is not[53]. We included accuracy, precision, recall, specificity, and F1 score as performance metrics for both training and testing (detailed in the Supplementary Information).

To investigate the effect of amino acid substitutions on both yield and glycan binding phenotype, we employed a grouping method for amino acids[54]. Each amino acid was assigned to one of three groups based on its biophysical properties: nonpolar (V, L, I, M, C, F, W, and Y), small polar (G, A, and P), and polar/charged (S, T, N, Q, H, D, E, K, and R). HA protein sequence was encoded into a vector $X_i$ by comparing to a wild-type sequence and if a mutation occurred in residue $j$ (e.g., nonpolar to small polar), we encoded the $j$-th element of $X_i$ to 1; otherwise, we encoded it to 0. To evaluate the directionality of amino acid substitutions on both yield and glycan binding phenotype, we used three different sparse models (LASSO, RIDGE, and SGL) and performed bootstrap analyses (detailed in Supplementary Data 5–7). In brief, we selected all features with a bootstrap value cutoff of 80 from 100 independent runs.

### Predictive model

In this study, a predictive model was developed to estimate the antigenic distance between two viruses based on their genetic sequences. The model was defined as follows:

$$\hat{y} = x\left(\mu w^{global} + (1-\mu)w^{local}\right), \qquad (7)$$

where x is the genetic distance vector between the two viruses, $\hat{y}$ is the predicted antigenic distance between them, $w^{global}$ is the global weight representing the average of weights across different tasks, $w^{local}$ indicates the weights from each individual task, and $\mu$ is set to 0.4 to balance the global and local weights.

In addition, a scoring function was proposed to measure yield differences between two viruses based on their amino acid sequences. The scoring function is defined as follows:

$$\hat{y} = \sum_{k=1}^{K} \sum_{i_1,\cdots,i_k}^{d} w^{(k)}_{<i_1,\cdots,i_k>} z^{(k)}_{<i_1,\cdots,i_k>}, \qquad (8)$$

Here, w and z were the weight and feature matrices used in the GHSM approach mentioned above. The detailed prediction results for both the antigenic distance and yield differences are presented in Supplementary Data (Supplementary Data 12).

### Cells and viruses

Human embryonic kidney (293T) cells and Madin-Darby canine kidney (MDCK) CCL-34 cells were obtained from the American Type Culture Collection (Manassas, VA). The cells were maintained in Dulbecco's Modified Eagle Medium (GIBCO/BRL, Grand Island, NY; catalog number 11965-092) supplemented with 5% fetal bovine serum (Atlanta Biologicals, Lawrenceville, GA; catalog number S12450H) and penicillin-streptomycin (Invitrogen, Carlsbad, CA; catalog number 15140122) at 37 °C with 5% CO2. The HA gene of CA/04 was cloned into the vector pHW2000 and used as a template to construct the mutant library. The viruses generated by reverse genetics were propagated in MDCK cells and cultured at 37 °C with 5% CO2 in Opti-MEM medium (GIBCO/BRL, Grand Island, NY; catalog number 11058-021) supplemented with 1 μg/ml of TPCK (N-tosyl-L-phenylalanine chloromethyl ketone)-Trypsin (Sigma-Aldrich, St. Louis, MO; catalog number T1426) and penicillin-streptomycin (Invitrogen, Carlsbad, CA; catalog number 15140122). The virus titers were determined by TCID$_{50}$ in MDCK cells.

### Sequence and serological Data

Serologic data for A(H1N1) viruses were collected from data described elsewhere[13,55,56], and included HAI titers generated between 1,015 viruses and 194 serum samples (Supplementary Data 15). A total of 11,424 A(H1N1)pdm09 HA protein sequences from 2009 to 2020 were obtained from GISAID (https://gisaid.org).

### Construction of plasmid library, gene synthesis, and rescue of mutants

The mutant plasmid library with random mutations in the HA RBS was generated using the epPCR strategy[15]. Four primers were used to generate the HA-pHW2000 RBS mutant library: 1) 130loop_F: 5′-TCA TGG CCC AAT CAT GAC TCG AAC-3′; 2) 190helix_F: 5′-TGG GGC ATT CAC CAT CCA TCT ACT-3′; 3) 190helix_R: 5′-AAC ATA TGT ATC TGC ATT CTG ATA-3′; and 4) 220loop_R: 5′-TAG TGT CCA GTA ATA GTT CAT TCT-3′. The epPCR product (2 μl) was transfected into XL1-Blue Supercompetent Cells (Agilent Technologies, Santa Clara, CA; catalog number 200236). The transformed cells were directly inoculated onto LB (Luria Bertani) agar plates, and the clones were propagated in 5 ml of LB media. The clones generated from the RBS mutant library were confirmed by Sanger sequencing using the sequencing primer 5′-GAA

CGT GTT ACC CAG GAG ATT-3′. Mutant viruses were rescued by plasmid-based reverse genetics with the NA genes from CA/04 and six internal genes from PR8, as described elsewhere[57]. To compare the phenotypes of the predicted vaccine candidates, we also generated the WT virus, a parent prototype 6:2 reassortant virus ith wild-type HA and NA genes from CA/04 and six internal genes from PR8, by using reverse genetics. To confirm the lack of any undesired egg or cell-adapted amino acid changes, each mutant's HA genes were confirmed by using Sanger sequencing post-rescue and propagation.

To validate the antigenic and high-yield properties of the viruses predicted by our computational model, we synthesized the HA and NA genes for four MAIVeSS-predicted potential vaccine viruses from epidemic strains (Gene Universal Inc., Newark, DE) and then generated reassortant viruses with the HA and NA from each of these testing epidemic strains and the six internal genes from PR8 using reverse genetics: A/Saint-Petersburg/RII57/2016(H1N1) (HA,NA)×PR8(rgSP/16), A/Chongqing-Yuzhong/SWL1453/2017(H1N1)(HA,NA)×PR8(rgCQ/17), A/Brunei/25/2019(H1N1)(HA,NA)×PR8(rgBRU/19), and A/Malaysia/33075487/2020(H1N1)(HA,NA)×PR8(rgMAS/20).

## Evaluation of viral yield

To evaluate the effect of mutations on viral yield, we performed cell culture assays and embryonated egg assays. For the cell culture assays, we inoculated MDCK cells with each influenza virus at a multiplicity of infection of 0.001 and incubated the cells at 37 °C with 5% CO2 for 1 h. After incubation, the inocula were removed, and the cells were washed twice with phosphate-buffered saline (PBS). Then, the cells were incubated with Opti-MEM I (GIBCO, Grand Island, NY; catalog number 11058-021) containing TPCK-trypsin (Sigma-Aldrich, St. Louis, MO; catalog number T1426) (1 μg/ml) at 37 °C with 5% CO2. After 48 h, 200 μl of supernatants were collected, aliquoted, and stored at −80 °C until use. For the embryonated egg assays, 9-day-old specific pathogen-free chicken eggs were inoculated with 200 $TCID_{50}$ of each virus and incubated at 37 °C for 72 h, and allantoic fluid were collected. The viral titers in the samples from both the MDCK cells and the embryonated eggs were determined using TCID50 assays in MDCK cells.

Conventional methods for quantifying yields of inactivated influenza vaccines rely on the HA protein, typically determined by SDS-PAGE gel following PNGase treatment. However, this procedure is labor-intensive, preventing us from quantifying the yields of all 196 mutants propagated in both eggs and cells. We quantified the total proteins obtained from ultracentrifugation purification of supernatants from virus-infected cell or egg cultures and analyzed their correlations with the viral titration TCID50. Pearson correlation analysis showed that the total protein yields are positively correlated with TCID50 (Supplementary Fig. 14). Additional SDS-PAGE analyses indicated that approximately 40% of the total proteins are HA proteins (Supplementary Figs. 15 and 16). Consequently, in this study, we used TCID50 titers to assess vaccine yield.

## Virus concentration and purification

Viruses for the glycan microarray analysis were purified as described elsewhere[23]. Briefly, viruses were purified from the cell supernatant or allantoic fluid by low-speed clarification (2482 × g, 30 min, 4 °C) to remove debris and then followed by ultracentrifugation through a cushion of 30–60% sucrose in a 70Ti Rotor (Beckman Coulter, Fullerton, CA) (100,000 × g, 3 h, 4 °C). The virus pellet was re-suspended in 100 μl of PBS and stored at −80 °C until use.

## Glycan microarray

To identify unique substructures bound specific sets of mutants, a glycan microarray with 75 glycoforms were printed on *N*-hydroxysuccinimide (NHS)–derivatized slides[23]. The 75 glycans were selected to represent four different glycan categories, including N-glycans,

Asn-linked N-glycans, Gangliosides, Thr-linked O-mannosyl glycans (Supplementary Fig. 3). These glycans on the microarray have the same base structures and spacer arms but different terminal structures. The glycans were printed in replicates of four in a subarray, and sixteen subarrays were printed on each glass slide. All glycans were prepared at a concentration of 100 mM in phosphate buffer (100 mM sodium phosphate buffer, pH 8.5). The slides were fitted with a 16-chamber adapter to separate the subarrays into individual wells for assay. The unreacted NHS groups on the slides are blocked with 50 mM ethanolamine in 50 mM sodium borate buffer (pH 9.2) at 4 °C for 1 h and then the slides are rinsed with water. Before the assay, slides were rehydrated for 5 min in TSMW buffer (20 mM Tris-HCl, 150 mM NaCl, 0.2 mM $CaCl_2$, and 0.2 mM $MgCl_2$, 0.05% Tween). Viruses are purified by sucrose density gradient ultracentrifugation and titrated to about 32,000 hemagglutination units/ml. Then 10 μl of 1.0 M sodium bicarbonate (pH 9.0) was added to 80 μl of virus, and the virus was incubated with 10 μg of Alexa Fluor 488 NHS Esters (Succinimidyl Esters; Invitrogen, Carlsbad, CA; catalog number A20100) for 1 h at 25 °C. After overnight dialysis to remove excess Alexa 488, viruses HA titer were checked and then bound to glycan array. Labeled viruses were incubated on the slide at 4 °C for 2 h, washed, and centrifuged briefly before being scanned with an InnoScan 1100 AL fluorescence imager (Innopsys, Carbonne, France).

## Haemagglutination and HAI assays

Haemagglutination and HAI assays were performed by using 0.5% turkey erythrocytes as described by the WHO Global Influenza Surveillance Network Manual for the Laboratory Diagnosis and Virological Surveillance of Influenza. Turkey erythrocytes were obtained from Lampire Biological Products (Everett, PA; catalog number 7209403). The turkey erythrocytes were washed three times with 1 × PBS (pH 7.2) before use and then diluted to 0.5% in 1 × PBS (pH 7.2). Ferret antisera used in the HAI assays were produced by infecting influenza seronegative ferrets (see details in Supplementary Information) or obtained from BEI Resources (https://www.beiresources.org) or International Reagent Resource (https://www.internationalreagentresource.org).

## Biolayer interferometry assays (BLI)

The virus receptor binding affinities were determined by BLI with an Octet RED instrument (Pall ForteBio, Menlo Park, CA). Five biotinylated glycan analogs, Neu5Acα2-3Galβ1-4-GlcNacβ-PAA-biotin (3′SLN) (Lectinity Holdings, Moscow, Russia; catalog number 0036-BP), Neu5Acα2-6Galβ1-4GlcNacβ-PAA-biotin (6′SLN) (Lectinity Holdings, Moscow, Russia; catalog number 0997-BP), Neu5Acα2-3Galβ1-4(Fucβ1-3)GlcNacβ-PAA-biotin (sLe^X), Neu5Gcα2-3Galβ1-4GlcNAcβ-PAA-biotin (3′SLN(Gc)), or Neu5Gcα2-3Galβ1-4(Fucβ1-3]GlcNAcβ-PAA-biotin (SLe^{X(Gc)})] were used. Among them, SLe^X, 3′SLN(Gc), and SLe^{X(Gc)} were synthesized. The glycans were preloaded onto streptavidin-coated biosensors at up to 0.3 μg/ml for 5 min in 1 × kinetic buffer (Pall FortéBio, Menlo Park, CA; catalog number 18-1092). Each test virus was diluted to a final concentration of 100 pM with 1 × kinetic buffer containing 10 μM oseltamivir carboxylate (American Radiolabeled Chemicals, St. Louis, MO; catalog number HY-13318) and zanamivir (Sigma-Aldrich, St. Louis, MO; catalog number SML0492-10MG) to prevent cleavage of the receptor analogs by NA proteins of virus. Association was measured for 30 min at 25 °C. Responses were normalized by the highest value obtained during the experiment, and binding curves were fitted by using the binding-saturation method in GraphPad Prism 8 (https://www.graphpad.com/scientific-software/prism/). The normalized response curves report the fractional saturation ($f$) of the sensor surface as described in elsewhere[58]. The $RSL_{0.5}$ values were calculated to determine the binding affinity between a virus and glycan analog pair, using the binding-saturation method in GraphPad Prism 8 software. Higher $RSL_{0.5}$ values indicate weaker binding affinity between the virus and glycan analog.

**Table 3 | Hemagglutination inhibition (HAI) titers of the vaccinate candidates selected by MAIVeSS for 2009 H1N1 viruses**

| Viruses | Ferret antisera | | | | |
|---|---|---|---|---|---|
| | CA/04 | CA/07 | UT/09 | MI/15 | WI/19 |
| A/California/04/2009(H1N1pdm) (CA/04) | **320.00** | 640.00 | 320.00 | 320.00 | 40.00 |
| A/California/07/2009(H1N1pdm) (CA/07) | 640.00 | **640.00** | 320.00 | 640.00 | 40.00 |
| A/Utah/20/2009(H1N1)pdm09 (UT/09) | 1280.00 | 160.00 | **1280.00** | 160.00 | 80.00 |
| A/Michigan/45/2015(HA,NA) (MI/15) | 640.00 | 1280.00 | 640.00 | **1280.00** | 80.00 |
| A/Wisconsin/588/2019(H1N1) (WI/19) | 20.00 | 20.00 | 80.00 | 80.00 | **2560.00** |
| A/Saint-Petersburg/RII57/2016 (H1N1)(HA,NA)xPR8 (rgSP/16) | 640 | 640 | 320 | 640 | 40 |
| A/Chongqing-Yuzhong/SWL1453/2017(H1N1)(HA,NA)xPR8 (rgCQ/17) | 160 | 80 | 320 | 160 | 40 |
| A/Brunei/25/2019(H1N1)(HA,NA)xPR8 (rgBRU/17) | 10.00 | 10.00 | 10.00 | 20.00 | 160.00 |
| A/Malaysia/33075487/2020(H1N1)(HA,NA)xPR8 (rgMAS/17) | 20.00 | 10.00 | 80.00 | 40.00 | 1280.00 |

The HAI assays were performed in triplicate using 0.5% turkey red blood cells. The homologous HAI titers are highlighted in bold. Ferret antisera were produced by infecting influenza seronegative ferrets (see details in Supplementary Information) or obtained from BEI Resources (https://www.beiresources.org) or International Reagent Resource (https://www.internationalreagentresource.org).

## Structural modeling and visualization of proteins structure

The three-dimensional structure of HA protein was modeled based on the crystal structure of CA/04 HA in complex with 6'SLN (PDB ID# 3UBN) and 3'SLN (PDB ID# 3UBQ). Coot (https://www2.mrc-lmb.cam.ac.uk/personal/pemsley/coot/) was used to introduce the desired mutation to the three subunits of a HA trimer. The mutated coordinates were subsequently refined by energy minimization using Phenix (https://phenix-online.org). Structure figures were made using Pymol (The PyMOL Molecular Graphics System, Version 1.3, Schrödinger, LLC).

## Statistics and reproducibility

To minimize the risk of overfitting and ensure a reliable assessment of our model's performance, we randomly assigned 90% of our dataset for training and validation, reserving the remaining 10% exclusively for testing. Importantly, the testing set remained untouched during parameter tuning to maintain an impartial evaluation of the model's generalization capabilities. Pearson correlation coefficient was used to measure the model performance and the association between the total protein yields and TCID50 for a testing influenza virus. The model performance matrices include accuracy, specificity/recall, sensitivity, precision, and F1 score (see details in Supplementary Information). Bootstrap experiments were used to evaluate the robustness of the model in feature selection. A total of 100 independent bootstrapping experiments were conducted, with each experiment randomly selecting 80% of the data. The frequency of a feature's selection across these experiments was used as an indicator of its importance, and the higher the frequency, the more important the feature. No statistical method was used to predetermine sample size, and no data were excluded from the analyses. The Investigators were not blinded to allocation during experiments and outcome assessment.

## Reporting summary

Further information on research design is available in the Nature Portfolio Reporting Summary linked to this article.

## Data availability

All relevant data supporting the key findings of this study are available within the article and its Supplementary Information files. The serological data for the vaccinate candidates generated in this study are available in Table 3. The list of the wild type CA/04 and the HA RBS mutant viruses and their associated antigenicity, virus yield in egg and cells, and glycan binding properties generated in this study are available at Supplementary Data 4. The serological data from public sources[13,55,56], including HAI titers generated between 1,015 viruses and 194 serum samples (Supplementary Data 15). The glycans printed on microarray array are available in Supplementary Fig. 1. The GISAID accession numbers for the epidemic A(H1N1)pdm09 strains are available from Supplementary Data 12. To access the GISAID database (https://gisaid.org), users need to log in following the instructions provided by the GISAID database. Once logged in, the GISAID database enables users to search and retrieve sequence and metadata data using either a specific accession number or a specific strain name. The three-dimensional structure of the HA protein was modeled by referencing the crystal structures of CA/04 HA (PDB ID# 3LZG) in complex with 6' SLN (PDB ID# 3UBN) and 3 SLN (PDB ID# 3UBQ). Additionally, the original data utilized for generating bar graphs and geospatial visualizations can be accessed in the Source Data file. Source data are provided with this paper.

## Code availability

The source codes for the model development in this study can be accessed through GitHub at https://github.com/FluSysBio/MAIVeSS and also through Code Ocean at https://doi.org/10.24433/CO.8910619.v1[59]. The MAIVeSS webserver can be accessed at http://sysbio.missouri.edu/software/MAIVeSS.

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

## Acknowledgements

We would like to acknowledge Feng Chen, George Eamon Sarafianos, Xin Alice Liu for their assistance in sample processing, virus propagation, and serological assays. We also gratefully acknowledge the originating laboratories responsible for obtaining the specimens and the submitting laboratories where genetic sequence data were generated and shared via the GISAID Initiative, which formed the basis for this research. We thank Jian Zhang at Z Biotech (www.zbiotech.com) for array printing. This project was supported by grants from the US National Institutes of Health (1R01AI116744, R01AI147640, and R21AI144433) and Welch Foundation (C-1565 to Y.J.T.). Cynthia Tang was also partially supported by a grant from the US National Institutes of Health (F30AI172230).

## Author contributions

Conceptualization: X.F.W.; Methodology: C.G., F.W., X.F.W.; Investigation: C.G., F.W., M.G., L.L., B.P., and X.F.W.; Visualization: C.G., B.H., C.Y.T., J.Z., Y.J.T., and X.F.W.; Funding acquisition: X.F.W., Y.J.T.; Project administration: X.F.W.; Supervision: X.F.W.; Writing, original draft: C.G. and X.F.W.; Writing, review and editing: C.G., F.W., M.G., L.L., B.P., C.Y.T., J.Z., F.L., H.X., R.W., Y.J.T., and X.F.W

## Competing interests

The authors declare no competing interests.
