## [Peer Review File · Nature Communications]

MAIVeSS: Streamlined selection of antigenically matched, high-yield viruses for seasonal influenza vaccine productionReviewer #1 (Remarks to the Author):

In this article the authors presented a Machine-learning Assisted Influenza VaccinE Strain Selection (MAIVeSS) framework that was trained to rapidly identify viruses suitable as candidate influenza vaccine viruses. MAIVeSS was designed to learn genetic features associated with antigenicity, growth ability and receptor binding, using a big data of a random mutant virus library.

The 189 random mutated viruses had one to seven random mutations within or near the receptor binding site of hemagglutinin (HA) from A/California/04/2009 (CA/04) (H1N1pdm09). The mutant viruses were generated using plasmid-based reverse genetics with the NA gene of CA/04 and six internal genes from A/Puerto Rico/8/34. Rescued mutant viruses were analyzed for growth, antigenicity and receptor binding properties. The authors proposed that applying MAIVeSS to publicly available sequences would allow the identification of antigenically matched, high-yield viruses among circulating viruses in a short time.

The great amount of experiments and calculations the authors have done to demonstrate the usefulness of MAIVeSS is commendable.

Inactivated egg-based vaccine is currently the most commonly used influenza vaccine in the world. However, it is well known that amino acid substitutions occur in HA during adaptation of the virus to embryonated chicken eggs, resulting in changes in the antigenicity. In addition, reassortment between high-growth master viruses and wild type viruses do not always result in improvement in the yield. To solve these problems the authors proposed using MAIVeSS to select high-yielding strains that do not undergo antigenic mutation from publicly available sequences of the circulating influenza viruses.

However, MAIVeSS was designed to learn the genetic features derived from mutant viruses rescued in 293T and MDCK cells. Considering that seasonal inactivated egg-based vaccine viruses have been developed by inoculating clinical samples directly into embryonated chicken eggs and passaging only in the eggs, the effect of virus passaging in the eggs on the amino acid sequences of HA need to be considered, but this effect was not considered in this study.

Therefore, it cannot be judged that the MAIVeSS is useful for their purpose of selecting antigenically matched and high-yield egg-based influenza vaccine virus from clinical samples.

Regarding the selection of high-yield vaccine viruses, as described below, the concept of 'yield' in vaccine manufacturing differs from that described in the manuscript. Therefore it is unlikely that MAIVeSS is helpful for selecting high-yield vaccine viruses.

As for Mutant library:

The established mutant library consists of the viruses with mutations in the receptor binding site (RBS) of CA/04-HA. However, mutations in the circulating influenza virus HA do not occur exclusively within or near the RBS. The mutations then accumulate in HA each year, resulting in gradual changes in structure and antigenicity of HA. In addition, mutations also accumulate in the NA and internal genes of the circulating influenza viruses. HA-NA interactions have been reported to affect receptor binding ability and specificity. It is also well-known that the compatibility between gene segments also influences virus replication. Therefore it is doubtful whether MAIVeSS established with the random mutant library will be helpful in selecting suitable candidate vaccine viruses from among upcoming viruses.

Regarding 'yield' traits:

When considering vaccine production, the 'yield' of the inactivated vaccine is estimated from the amount of HA antigen per fixed amounts of infected eggs or cell culture, instead of from the infectivity of the vaccine virus. Because manufactured inactivated influenza vaccine needs to contain a defined amount of HA antigen. To select high-yield vaccine viruses, the 'amount of HA antigen' of the viruses need to be measured. Because 'HA yield' is one of the key considerations in the current CVV (candidate vaccine virus) selection process. TCID50 titers of the virus is not helpful to select egg-based vaccine virus.

Contradictory statements

There are contradictory statements about the receptor binding properties of WI/19 as follows. Corrections are required.

p.9, lines 211-214

The results showed that 3 of the MAIVeSS predicted CCVs, rgSP/16, rgCQ/17 and rgMAS/20, bound to both 3'SLN and 6'SLN, whereas MI/15, WI/19, AND ONE PREDICTED CCV, RGRU/19, BOUND ONLY TO 6'SLN. Furthermore, we found that rgBRU/19 had a higher binding avidity to 6'SLN than MI/15 (Fig. 5A).

p.11, lines 261-263

On the other hand, some high-yield strains (e.g. WI/19) WERE FOUND TO HAVE NO SIGNIFICANT DIFFERENCES IN THEIR BINDING PREFERENCES FOR THE 3'SLN OR 6'SLN COMPARED to CA/04 and MI/15, as tested by our experiments (Fig. 5A),

p.30, lines 705-707

Results showed that rgSP/16, rgCQ/17, and rgMAS/20 bound to both 3'SLN and 6'SLN, whereas MI/15, rgBRU/19, and WI/19 BOUND EXCLUSIVELY TO 6'SLN.

Clearer explanation is needed:

Following sentences are unintelligible. Is the authors' conclusion that the vaccine viruses can be selected from viruses prevalent in nature without further engineering? However, even now, no special engineering is done. Do they recommend using the wild type virus as a vaccine virus, instead of a reassortant virus?

Clearer explanations is needed.

p.12 lines 273-278

In humans, there is no direct selection pressure to increase cell-based or egg-based replication efficiency.

Thus, our findings suggested that ad hoc substitutions at the HA RBS across A(H1N1)pdm09 viruses likely enabled a subset of these variants to expand their binding preference from SA2-6Gal to both SA2-6Gal and SA2-3Gal, resulting in the acquisition of a high-yield trait. This study demonstrates that it is possible to select naturally circulating strains as vaccine candidates without the need for further engineering.

Minor comments:

1. For Table S4, HAI titers of some mutants are missing (R017, R042 etc.). It is necessary to state the reason why the HAI titers are not shown.
2. It is helpful to understand the contents if the amino acid numbering for A/H1N1pdm09 HA is used instead of that for A/H3N2. Amino acid numbering and antigenic regions of A/H1N1pdm09 HA have already been used in other papers. This will make it easier to compare the results of this manuscript with those of other papers.
3. For Table S12, amino acid numbers after 99 are shown as ##. The actual number for each residue should be displayed.
4. P.6 lines 126-127
The authors used MDCK cells to measure the infectious titers of the egg-grown virus. However, when the egg-grown virus is to be used for the production of egg vaccines, the egg infectious dose titer (EID50) is required.
5. What does it mean by 'CCV'? Do you mean 'CVV'?

Reviewer #2 (Remarks to the Author):

This paper addresses an important problem using appropriate methods.

-> What are the noteworthy results?

The proposed machine learning/data analysis framework can potentially reduce the optimal candidate vaccine virus selection time from months to days and thus facilitate timely supply of seasonal vaccines.

-> Will the work be of significance to the field and related fields?

The problem addressed in this paper is of significance to the field. However, I have concerns about the results in this specific paper. I mention my concerns below.

-> How does it compare to the established literature? If the work is not original, please provide relevant references.

The authors don't compare their results to the established literature. This is very concerning because there are many studies on this exact same problem in the literature. It also seems that all the methods used in this paper have been developed in other papers and already used in a similar context in other published papers e.g. [20,31,39]. I am not sure about the main contribution of this paper given these other studies.

-> Does the work support the conclusions and claims, or is additional evidence needed?

There is a lot of data presented in the paper to support the discussion. However, the presentation is not organized. In most cases it is not possible to follow the origin of the evidence presented because there is not a coherent discussion. The authors jump from one topic to another and keep reporting different data elements.

-> Are there any flaws in the data analysis, interpretation and conclusions? Do these prohibit publication or require revision?

Yes, there are. Performance comparison, parameter optimization and bootstrapping analysis results are reported on a training set in Tables S1-S3 based on RMSE. The authors argued that low RMSE indicates high performance, but they never tested their models on an out-of-sample testing set. In this case, low RMSE can just mean overfitting. Results from a testing set must be reported.

-> Is the methodology sound? Does the work meet the expected standards in your field?

The methodology is sound but not novel. The methodologies used in this paper have been used to answer very similar questions in the literature.

-> Is there enough detail provided in the methods for the work to be reproduced?

There are some details but not enough to reproduce the results. The discussion of methods is very basic and lacks important details. I believe this is mainly because those methods have been developed in other studies.

Reviewer #3 (Remarks to the Author):

In the manuscript under consideration, the authors developed a machine learning-based method (named MAIVeSS) to identify influenza vaccine viruses with antigenic and high-yield phenotypes directly from sequences obtained from clinical samples, which would be ideal and could potentially accelerate vaccine production timelines. Then, they applied MAIVeSS on publicly available sequences to select A(H1N1)pdm09 CVVs and experimentally confirmed that these CVVs had optimal antigenicity and growth in cells and eggs. Although the manuscript is well organized, I have the following major concerns from the computational perspective.

1) The MAIVeSS framework developed by the authors attempted to identify influenza vaccine viruses with antigenic and high-yield phenotypes, which can be formulated as a multi-objective

optimization problem. It seems that MAIVeSS cannot directly generate mutations or mutation combinations which satisfy multiple objectives simultaneously, which are more useful in practical applications simultaneously.

2) The MAIVeSS framework lacks interpretation/visualization on what attributes in the genomic sequences associated with antigenic and high-yield phenotypes. The machine learning algorithm should automatically provide these relationships for the input sequences in the test case.

3) In Ln 85, the authors claimed that the same principles can be readily applied to other subtypes of influenza viruses. How to convince the readers about the generalization ability of the proposed principles? Is there web server available to potential users which could select influenza vaccine seeds for other subtypes?

4) In the recent publication (<https://doi.org/10.1038/s41467-023-39199-6>), a machine learning-guided antigenic evolution prediction (MLAEP), which combines structure modeling, multi-task learning, and genetic algorithms to predict the viral fitness landscape and explore antigenic evolution via in silico directed evolution. What are the advantages of MAIVeSS framework in comparison with MLAEP? Moreover, did the authors explore the power of language models when they developed MAIVeSS?

Responses to Reviewers' Comments

Reviewer #1 (Remarks to the Author):

In this article the authors presented a Machine-learning Assisted Influenza Vaccine Strain Selection (MAIVeSS) framework that was trained to rapidly identify viruses suitable as candidate influenza vaccine viruses. MAIVeSS was designed to learn genetic features associated with antigenicity, growth ability and receptor binding, using a big data of a random mutant virus library.

The 189 random mutated viruses had one to seven random mutations within or near the receptor binding site of hemagglutinin (HA) from A/California/04/2009 (CA/04) (H1N1pdm09). The mutant viruses were generated using plasmid-based reverse genetics with the NA gene of CA/04 and six internal genes from A/Puerto Rico/8/34. Rescued mutant viruses were analyzed for growth, antigenicity and receptor binding properties. The authors proposed that applying MAIVeSS to publicly available sequences would allow the identification of antigenically matched, high-yield viruses among circulating viruses in a short time. The great amount of experiments and calculations the authors have done to demonstrate the usefulness of MAIVeSS is commendable.

Response: Many thanks for your compliment!

Inactivated egg-based vaccine is currently the most commonly used influenza vaccine in the world. However, it is well known that amino acid substitutions occur in HA during adaptation of the virus to embryonated chicken eggs, resulting in changes in the antigenicity. In addition, reassortment between high-growth master viruses and wild type viruses do not always result in improvement in the yield. To solve these problems the authors proposed using MAIVeSS to select high-yielding strains that do not undergo antigenic mutation from publicly available sequences of the circulating influenza viruses.

However, MAIVeSS was designed to learn the genetic features derived from mutant viruses rescued in 293T and MDCK cells. Considering that seasonal inactivated egg-based vaccine viruses have been developed by inoculating clinical samples directly into embryonated chicken eggs and passaging only in the eggs, the effect of virus passaging in the eggs on the amino acid sequences of HA need to be considered, but this effect was not considered in this study.

Therefore, it cannot be judged that the MAIVeSS is useful for their purpose of selecting antigenically matched and high-yield egg-based influenza vaccine virus from clinical samples.

Response: Thank you for your insightful comments. Indeed, amino acid substitutions in culture-adapted strains pose challenges during the selection process for traditional isolate-based, particularly egg-isolate-based approaches. The primary objective of our proposed method is to streamline the vaccine strain selection from clinical samples, eliminating the need for virus propagation and, consequently, bypassing the issues with egg-based isolates. In this study, we produced mutants in 293T/MDCK cells by using established plasmid-based reverse genetics techniques, and then propagated these mutants in SPF chicken eggs or MDCK cells. To confirm the lack of any unexpected egg or cell-adapted amino acid changes, each mutant's HA genes were confirmed by using Sanger sequencing post-rescue and propagation, and this has been clarified in the updated manuscript. Incorporating historical data, our model identified specific amino acid changes that can influence the H1N1 viruses' antigenicity, and, in the updated manuscript, we now highlighted those identified in prior studies as indicative of egg adapted substitutions (Tables 1 and 2). Specifically, amino acid changes Y/H17Y, A57V, D131E, , R193S, L194I, V223M, D225D/G, Q226R, G228G/A, of 2009 H1N1 viruses were identified by comparing cell and egg derived isolates, and these residues are suggested to be potential egg adaption; of them, D127E, L191I, D225G/N and Q226R can affect antigenic properties (Yang et al. 2019).

Although our model was intended to be primarily designed for sequences derived from clinical samples, it's adaptable to sequences sourced from isolates, including those from eggs. As specified in Line 184, our model does predict the viruses present in 897 clinical samples are with high growth traits: "Of the 1,349 viruses identified as high-yield variants, 897 had HA sequences that were directly generated from clinical swabs, while the remaining sequences were generated from viruses grown either in cells (n = 331) and eggs (n = 83). The virus source for HA sequence was unclear for the remaining 38 high-yield variants." Of note, for the four HA sequences we chose – three cell-based isolates A/Saint-Petersburg/RII57/2016(H1N1), A/Chongqing-Yuzhong/SWL1453/2017(H1N1), A/Brunei/25/2019(H1N1), and one egg-adapted isolate A/Malaysia/33075487/2020(H1N1) in the validation. Among these four strains, only egg-adapted A/Saint-Petersburg/RII57/2016(H1N1) has D225G, one potential egg adapted amino acid substitution reported by others (Yang et al. 2019). Nevertheless, these data support MAIVeSS's capability in predicting high-yield antigenic variants directly from clinical sample genomic sequences.

Yang, L. et al. Mutations associated with egg adaptation of influenza A (H1N1) pdm09 virus in laboratory based surveillance in China, 2009–2016. *1*, 41-45 (2019).

Regarding the selection of high-yield vaccine viruses, as described below, the concept of 'yield' in vaccine manufacturing differs from that described in the manuscript. Therefore it is unlikely that MAIVeSS is helpful for selecting high-yield vaccine viruses.

Response: Thank you for your great comments. Indeed, the quantities of the antigen in inactivated influenza vaccines are typically calculated based on the amounts of HA proteins. Conventional methods for quantifying yields of inactivated influenza vaccines rely on the HA protein, typically determined by SDS-PAGE gel following PNGase treatment. However, this procedure is labor-intensive, preventing us from quantifying the yields of all 196 mutants propagated in both eggs and cells. To ensure high-throughput needs, we utilized the MDCK cell-based TCID50 methods. We acknowledge this limitation and have provided clarifications in the revised manuscript (Line 495-502).

Fig. 1. Quantification of HA proteins from egg-grown A/California/04/2009(HA,NA) × A/Puerto Rico/8/1934(H1N1). Ultracentrifugation products totaling 5 ug were treated with 500 units of PNGase F, followed by analysis using 12% SDS-PAGE.

The HA protein usually constitutes about 40% of the ultracentrifugation products from the supernatants of egg or cell growth, though there are variations. As a supporting data, we tested it for egg-grown A/California/04/2009(HA,NA) × PR8 (H1N1), which is the wild type virus we used in our study. In the updated manuscript, we updated the association analyses with TCID50 with the viral total proteins (Supplementary Fig. 14), which showed that the total viral proteins and TCID50 correlated well for both cell and egg based isolates.

Figure 2. Correlation between viral titration TCID50 in and the total proteins obtained from ultracentrifugation purification of supernatants from virus-infected cell or egg cultures. Pearson correlation analysis was conducted using Prism. Typically, a coefficient r between 0.4 and 0.7 is considered to indicate a moderate positive correlation, while a value above 0.7 signifies a strong positive correlation.

As for Mutant library:

The established mutant library consists of the viruses with mutations in the receptor binding site (RBS) of CA/04-HA. However, mutations in the circulating influenza virus HA do not occur exclusively within or near the RBS. The mutations then accumulate in HA each year, resulting in gradual changes in structure and antigenicity of HA. In addition, mutations also accumulate in the NA and internal genes of the circulating influenza viruses. HA-NA interactions have been reported to affect receptor binding ability and specificity. It is also well-known that the compatibility between gene segments also influences virus replication. Therefore it is doubtful whether MAIVeSS established with the random mutant library will be helpful in selecting suitable candidate vaccine viruses from among upcoming viruses.

Response: Thank you for your insightful comments. It is indeed true that the majority of mutations often lead to antigenic changes as a result of the selective pressure of herd immunity within human populations. However, we did note that a subset of viruses may carry random mutations on or adjacent to the HA receptor binding sites, albeit these strains may not be predominant. For instance, with 11,424 H1N1 viruses from public databases, most strains exhibited amino acid substitutions in the regions included in our random mutant library, and 730 even within the receptor binding sites (Supplementary Table 12). The existence of these viruses in naturally circulating strains provided us with a logical basis for identifying high-yield strains.

Regarding 'yield' traits:

When considering vaccine production, the 'yield' of the inactivated vaccine is estimated from the amount of HA antigen per fixed amounts of infected eggs or cell culture, instead of from the infectivity of the vaccine virus. Because manufactured inactivated influenza vaccine needs to contain a defined amount of HA antigen. To select high-yield vaccine viruses, the 'amount of HA antigen' of the viruses need to be measured. Because 'HA yield' is one of the key considerations in the current CVV (candidate vaccine virus) selection process. TCID50 titers of the virus is not helpful to select egg-based vaccine virus.

Response: Thank you for your great comments. Please refer to our earlier responses.

Contradictory statements

There are contradictory statements about the receptor binding properties of WI/19 as follows. Corrections are required.

p.9, lines 211-214

The results showed that 3 of the MAIVeSS predicted CCVs, rgSP/16, rgCQ/17 and rgMAS/20, bound to both 3'SLN and 6'SLN, whereas MI/15, WI/19, AND ONE PREDICTED CCV, RGBRU/19, BOUND ONLY TO 6'SLN. Furthermore, we found that rgBRU/19 had a higher binding avidity to 6'SLN than MI/15 (Fig. 5A).

p.11, lines 261-263

On the other hand, some high-yield strains (e.g. WI/19) WERE FOUND TO HAVE NO SIGNIFICANT DIFFERENCES IN THEIR BINDING PREFERENCES FOR THE 3'SLN OR 6'SLN COMPARED to CA/04 and MI/15, as tested by our experiments (Fig. 5A),

p.30, lines 705-707

Results showed that rgSP/16, rgCQ/17, and rgMAS/20 bound to both 3'SLN and 6'SLN, whereas MI/15, rgBRU/19, and WI/19 BOUND EXCLUSIVELY TO 6'SLN.

Response: Thank you! In the revised manuscript, we have revised the statement as “On the other hand, similar to CA/04 and MI/15, some high-yield strains (e.g. WI/19) bind only to 6'SLN but not 3'SLN” (Line 268-269).

Clearer explanation is needed:

Following sentences are unintelligible. Is the authors' conclusion that the vaccine viruses can be selected from viruses prevalent in nature without further engineering? However, even now, no special engineering is done. Do they recommend using the wild type virus as a vaccine virus, instead of a reassortant virus?

Clearer explanations is needed.

p.12 lines 273-278

In humans, there is no direct selection pressure to increase cell-based or egg-based replication efficiency. Thus, our findings suggested that ad hoc substitutions at the HA RBS across A(H1N1)pdm09 viruses likely enabled a subset of these variants to expand their binding preference from SA2-6Gal to both SA2-6Gal and SA2-3Gal, resulting in the acquisition of a high-yield trait. This study demonstrates that it is possible to select naturally circulating strains as vaccine candidates without the need for further engineering.

Response: Thank you for your comments. Indeed, the model of MAIVeSS intends to select HA sequences, which can be used to engineer a reassortant virus with an existing template (e.g. PR8). This has been validated through four engineering strains in our result section (Figure 4). Alternatively, the wild type virus with a high yield HA could be a target for evaluation as well. We have clarified this statement as “This study illustrates the feasibility of selecting HA sequences from naturally circulating strains as high yield candidates for recombinant vaccine development, by eliminating the need for further engineering” (Line 284-286).

Minor comments:

1. For Table S4, HAI titers of some mutants are missing (R017, R042 etc.). It is necessary to state the reason why the HAI titers are not shown.

Response: Thank you for your comments. Some mutants lost the ability to hemagglutinate turkey RBCs, precluding hemagglutination inhibition analyses. We have updated the notes in Supplementary Table 4 to explain.

2. It is helpful to understand the contents if the amino acid numbering for A/H1N1pdm09 HA is used instead of that for A/H3N2. Amino acid numbering and antigenic regions of A/H1N1pdm09 HA have already been used in other papers. This will make it easier to compare the results of this manuscript with those of other papers.

Response: Thank you. The amino acid numbering in this manuscript follows the convention for A/H3N2, as specified in Lines 97 and 108.

3. For Table S12, amino acid numbers after 99 are shown as ##. The actual number for each residue should be displayed.

Response: We appreciate your attention to detail. The error stemmed from visualization challenges, and we have since corrected it.

4. P.6 lines 126-127

The authors used MDCK cells to measure the infectious titers of the egg-grown virus. However, when the egg-grown virus is to be used for the production of egg vaccines, the egg infectious dose titer (EID50) is required.

Response: Thank you for your insightful comment. Indeed, performing EID50 for those viruses propagated in eggs would be ideal. However, the prohibitive cost of eggs (\$6.5 per SPF egg) deterred us from this approach. Still, as illustrated in Figure 2 shown earlier, the correlation between viral titration TCID50 and the total proteins derived from ultracentrifugation purification of supernatants from egg cultures was 0.8971. This suggests that the MDCK-based TCID50 can serve as a reliable indicator for virus titers propagated in eggs.

5. What does it mean by 'CCV'? Do you mean 'CVV'?

Response: Thank you again for your attention to detail. There was a typographical error, and the correct term should be "CVV". We have made this correction in the updated manuscript.

Reviewer #2 (Remarks to the Author):

This paper addresses an important problem using appropriate methods.

-> *What are the noteworthy results?*

The proposed machine learning/data analysis framework can potentially reduce the optimal candidate vaccine virus selection time from months to days and thus facilitate timely supply of seasonal vaccines.

-> *Will the work be of significance to the field and related fields?*

The problem addressed in this paper is of significance to the field. However, I have concerns about the results in this specific paper. I mention my concerns below.

Response: Thank you for your compliment! Please find below a point-by-point response to your concerns and comments.

-> *How does it compare to the established literature? If the work is not original, please provide relevant references.*

The authors don't compare their results to the established literature. This is very concerning because there are many studies on this exact same problem in the literature. It also seems that all the methods used in this paper have been developed in other papers and already used in a similar context in other published papers e.g. [20,31,39]. I am not sure about the main contribution of this paper given these other studies.

Response: Thank you for raising these concerns. Indeed, in the past nearly 15 years, my laboratory has focused on sequence-based vaccine strain selection. We selected sparse learning as the foundation for our problem and continued to improve the models based on the challenges we encountered. The three selected citations you mentioned indeed reflect some of the progress reports we had. We started with basic sparse learning (Cai et al. 2012), transitioned to multi-task sparse learning for overcoming data integration challenge (Han et al. 2019), and later incorporated multi-task group Lasso for overcoming feature type challenge (Li et al. 2019). The methods we used in this study are adapted from our previously published works, and we have provided detailed clarifications in the Online Methods section.

The objectives of this study are to integrate the best models we have developed, targeting both antigenicity and growth phenotypes, with the ultimate goal of providing a valuable tool for the public use. While we are very careful not to claim the machine learning methods are novel, we do believe the novelty include integrative toolkit, growth-centric studies, and the application of these tools in comprehending the antigenic evolution of H1N1 viruses. Beyond a new set of genetic markers associated with growth derived from our 196-mutant datasets, we also attempted to study the molecular mechanisms influencing high growth in contemporary H1N1pdm viruses. We believe, as an extension of our longstanding objective in sequence-based vaccine strain selection, this study can facilitate the selection of naturally circulating influenza vaccine strains with matching antigenicity and high-yield as seed viruses for influenza vaccine production. The integrated antigenic datasets and new growth data shared through this study can facilitate development of novel tools by the peers in these domains.

In this study, we delved into advanced parameter optimization to enhance feature selection performance. Specifically, for antigenicity-related multi-task group Lasso model, we formulated the problem as $\min_W \frac{1}{2} \|Y - XW\|_F^2 + \lambda_1 \sum_{j=1}^p \|W_j\|_1 + \lambda_2 \sum_{t=1}^k \sum_{l=1}^q \alpha_l \|W_{G_l,t}\|_2 + \lambda_3 \sum_{t=1}^k \sum_{l=1}^q \alpha_l \|W_{G_l,t}\|_1$

Within this formulation, we used incorporated three regularization terms. We defined $R_1(W) = \lambda_1 \sum_{j=1}^p \|W_j\|_1$, $R_2(W) = \lambda_2 \sum_{t=1}^k \sum_{l=1}^q \alpha_l \|W_{G_l,t}\|_2$, and $R_3(W) = \lambda_3 \sum_{t=1}^k \sum_{l=1}^q \alpha_l \|W_{G_l,t}\|_1$ where R_1 promotes task-wise sparsity for all the tasks while R_2 and R_3 drive feature group sparsity across varied tasks. Distinctively, our method diverges from that in Li et al. 2019, which did not incorporate R_3 . The

strength of l_1 -norm regularization lies in its ability to render the result matrix sparse. A majority of its components are zeroes, while the few remaining components pinpoint a concise subset of essential features. Incorporating R_3 regularization further refines our method, yielding even sparser results compared to Li et al. 2019. However, we are cautious not to overstate its novelty, while underscoring our endeavor to optimize the model.

Following your suggestion, in this revised manuscript, we expanded our model comparison beyond our previous evaluations. We now include a more comprehensive assessment of our method, encompassing three primary influenza antigenicity-related machine learning models mentioned in the literature, along with three deep learning approaches. Specifically, the conventional machine learning methods consist of Support Vector Machine (SVM) (Liao et al. 2008; Agor et al. 2018; Yao et al. 2012), Naïve Bayes (Du et al. 2012; Peng et al. 2017; Liu et al. 2023), and Random Forest (Zhou et al. 2018; Yao et al. 2017; Zeller et al. 2021). The deep learning methods include Gated Recurrent Unit (GRU) and two natural language models: Long Short-Term Memory (LSTM) and Transformer, as suggested by the third reviewer. As the source codes for these models from the literature are not publicly accessible, we utilized the machine learning models available in MATLAB package (R2023a) for comparison. The comparison results for cross-validation using the training data and performance evaluation using the testing dataset are provided in Supplementary Tables 1-3. Overall, our method demonstrated superior accuracy compared to previously reported results.

To ensure reproducibility and enhance the transparency of these comparisons, we have made the source codes for both our models and the codes we used for other models on GitHub. In addition, we have provided access to all datasets, which encompass the complete dataset, the segmented training/development datasets, and testing datasets. All these resources can be accessed at <https://github.com/FluSysBio/MAIVeSS>. The package also features a readme file that guides users on how to utilize the codes.

References

- Li, L. et al. Multi-task learning sparse group lasso: a method for quantifying antigenicity of influenza A(H1N1) virus using mutations and variations in glycosylation of Hemagglutinin. *BMC Bioinformatics* **21**, 182 (2020).
- Cai, Z. et al. Identifying antigenicity-associated sites in highly pathogenic H5N1 influenza virus hemagglutinin by using sparse learning. *Journal of molecular biology* **422**, 145-155 (2012).
- Han, L. et al. Graph-guided multi-task sparse learning model: a method for identifying antigenic variants of influenza A (H3N2) virus. *Bioinformatics* **35**, 77-87 % @ 1367-4803 (2019).
- Agor JK, Özaltın OY. Models for predicting the evolution of influenza to inform vaccine strain selection. *Hum Vaccin Immunother.* 2018 Mar 4;14(3):678-683. doi: 10.1080/21645515.2017.1423152. Epub 2018 Feb 12. PMID: 29337643; PMCID: PMC5861780.
- Liao YC, Lee MS, Ko CY, Hsiung CA. Bioinformatics models for predicting antigenic variants of influenza A/H3N2 virus. *Bioinformatics.* 2008 Feb 15;24(4):505-12. doi: 10.1093/bioinformatics/btm638. Epub 2008 Jan 10. PMID: 18187440.
- Yao B, Zhang L, Liang S, Zhang C (2012) SVMTriP: A Method to Predict Antigenic Epitopes Using Support Vector Machine to Integrate Tri-Peptide Similarity and Propensity. *PLOS ONE* 7(9): e45152. <https://doi.org/10.1371/journal.pone.0045152>

- Du X, Dong L, Lan Y, Peng Y, Wu A, Zhang Y, Huang W, Wang D, Wang M, Guo Y, Shu Y, Jiang T. Mapping of H3N2 influenza antigenic evolution in China reveals a strategy for vaccine strain recommendation. *Nat Commun.* 2012 Feb 28;3:709. doi: 10.1038/ncomms1710. PMID: 22426230.
- Peng Y, Wang D, Wang J, Li K, Tan Z, Shu Y, Jiang T. A universal computational model for predicting antigenic variants of influenza A virus based on conserved antigenic structures. *Sci Rep.* 2017 Feb 6;7:42051. doi: 10.1038/srep42051. PMID: 28165025; PMCID: PMC5292743.
- Liu M, Liu J, Song W, Peng Y, Ding X, Deng L, Jiang T. Development of PREDAC-H1pdm to model the antigenic evolution of influenza A/(H1N1) pdm09 viruses. *Virol Sin.* 2023 May 19:S1995-820X(23)00056-1. doi: 10.1016/j.virs.2023.05.008. Epub ahead of print. PMID: 37211247.
- Zhou X, Yin R, Kwok CK, Zheng J. A context-free encoding scheme of protein sequences for predicting antigenicity of diverse influenza A viruses. *BMC Genomics.* 2018 Dec 31;19(Suppl 10):936. doi: 10.1186/s12864-018-5282-9. PMID: 30598102; PMCID: PMC6311925.
- Yao Y, Li X, Liao B, Huang L, He P, Wang F, Yang J, Sun H, Zhao Y, Yang J. Predicting influenza antigenicity from Hemagglutinin sequence data based on a joint random forest method. *Sci Rep.* 2017 May 8;7(1):1545. doi: 10.1038/s41598-017-01699-z. PMID: 28484283; PMCID: PMC5431489.
- Zeller MA, Gauger PC, Arendsee ZW, Souza CK, Vincent AL, Anderson TK. Machine Learning Prediction and Experimental Validation of Antigenic Drift in H3 Influenza A Viruses in Swine. *mSphere.* 2021 Mar 17;6(2):e00920-20. doi: 10.1128/mSphere.00920-20. PMID: 33731472; PMCID: PMC8546707.

-> *Does the work support the conclusions and claims, or is additional evidence needed?*

There is a lot of data presented in the paper to support the discussion. However, the presentation is not organized. In most cases it is not possible to follow the origin of the evidence presented because there is not a coherent discussion. The authors jump from one topic to another and keep reporting different data elements.

Response: Thank you for your great comments. We added “To determine if the high-yield trait correlates with glycan substructure binding properties, we analyzed the receptor-binding properties of all 189 mutant viruses” to smooth the transition between amino acid feature discussion and glycan binding discussion (Line 262).

-> *Are there any flaws in the data analysis, interpretation and conclusions? Do these prohibit publication or require revision?*

Yes, there are. Performance comparison, parameter optimization and bootstrapping analysis results are reported on a training set in Tables S1-S3 based on RMSE. The authors argued that low RMSE indicates high performance, but they never tested their models on an out-of-sample testing set. In this case, low RMSE can just mean overfitting. Results from a testing set must be reported.

Response: Thank you for highlighting this concern. In our revised manuscript, we've provided a more comprehensive description of our data splitting procedure. We allocated 90% of our data for training and validation, and the remaining 10% for testing. Importantly, the test set was excluded from parameter optimization to avoid potential overfitting. Within the combined training and validation dataset, we further

segregated the data—90% for actual training and the remaining 10% for validation. We employed 10-fold cross-validation to fine-tune our parameters and evaluate the training performance.

To provide a more holistic view of our model's performance, we've expanded our evaluation metrics. In addition to accuracy, we now report recall, specificity, precision, and F1 scores. Clear definitions of each of these metrics have been added to Supplementary Information. Our updated Supplementary Table 1-3 illustrates both training and testing performance metrics. Consistency in these metrics across our models suggests minimal overfitting, indicating that our model generalizes well to new data.

We added two sections from Line 400 to 422 for clarification:

For antigenicity analyses, our comparison additionally included three primary machine learning models mentioned in the literature, along with three deep learning approaches. The conventional machine learning methods consist of Support Vector Machine (SVM),^{45, 46 47} Naïve Bayes,⁴⁸⁻⁵¹ and Random Forest.^{35, 52-54} The deep learning methods include Gated Recurrent Unit (GRU), Long Short-Term Memory (LSTM), and Transformer. As the source codes for these models from the literature are not publicly accessible, we utilized the machine learning models available in MATLAB package (R2023a) for comparison.

To develop and evaluate our models, we allocated 90% of our data for training and validation, and the remaining 10% for testing. The testing dataset was excluded from parameter optimization to avoid potential overfitting. Within the combined training and validation dataset, we employed 10-fold cross-validation by segregating the data, 90% for training and the remaining 10% for validation, to fine-tune our parameters and evaluate the training performance (Supplementary Figs. 8-13). As the results, we set λ_1 equals 2, λ_2 equals 0.01, and λ_3 equals 0.01 as optimal parameter for MTL-GGSL model, and λ equals 0.0001 and α equals 10 for GHSM model.

To evaluate the performance between models in MAIVeSS and the previously mentioned comparison models, for antigenicity analyses, we recast it as an antigenic distance classification problem to assess the model's efficacy in identifying antigenic variants. A virus pair is classified as an antigenic variant if its paired antigenic distance is 4-fold or greater; otherwise, it is not.⁵⁶ For yield analyses, we similarly formulated it as a classification problem: a virus is classified as a high yield virus if it has a ≥ 10 -fold increase in TCID₅₀/mL compared to the WT on the same substrate, otherwise, it is a low yield virus. We included accuracy, precision, recall, specificity, and F1 score as performance metrics (detailed in the Supplementary Information). For glycan binding analyses, we assessed their RMSE and Pearson correlation coefficient between predicted values and ground truth.

-> *Is the methodology sound? Does the work meet the expected standards in your field?*

The methodology is sound but not novel. The methodologies used in this paper have been used to answer very similar questions in the literature.

Response: Kindly refer to our previous responses.

-> *Is there enough detail provided in the methods for the work to be reproduced?*

There are some details but not enough to reproduce the results. The discussion of methods is very basic and lacks important details. I believe this is mainly because those methods have been developed in other studies.

Response: Please refer to our earlier responses for further details. To ensure reproducibility, we have provided access to all datasets, which encompass the complete dataset, the segmented training/development datasets, and testing datasets. Additionally, our original codes for MAIVeSS and the models we used for comparison are available. All these resources can be accessed at <https://github.com/FluSysBio/MAIVeSS>. The package also features a readme file that guides users on how to utilize the codes.

Reviewer #3 (Remarks to the Author):

In the manuscript under consideration, the authors developed a machine learning-based method (named MAIVeSS) to identify influenza vaccine viruses with antigenic and high-yield phenotypes directly from sequences obtained from clinical samples, which would be ideal and could potentially accelerate vaccine production timelines. Then, they applied MAIVeSS on publicly available sequences to select A(H1N1)pdm09 CVVs and experimentally confirmed that these CCVs had optimal antigenicity and growth in cells and eggs. Although the manuscript is well organized, I have the following major concerns from the computational perspective.

1) The MAIVeSS framework developed by the authors attempted to identify influenza vaccine viruses with antigenic and high-yield phenotypes, which can be formulated as a multi-objective optimization problem. It seems that MAIVeSS cannot directly generate mutations or mutation combinations which satisfy multiple objectives simultaneously, which are more useful in practical applications simultaneously.

Response: Thank you for your feedback. In this study, under the premise that high-yield antigenic variants can be discerned from naturally occurring strains, we developed MAIVeSS. This tool aims to streamline the selection of naturally circulating influenza vaccine strains that are both antigenically matched and have high yields directly from clinical samples. Our aspiration is that MAIVeSS can reduce the engineering steps needed for optimal candidate vaccine virus (CVV) preparation. Our model endeavors to pinpoint synergistic mutation pairs in residues closely located within the HA protein structure (please refer to Methods Line 382-385). While the genetic signatures we've identified for antigenicity and growth phenotypes might be valuable for laboratory engineering, that aspect remains outside the scope of this study.

2) The MAIVeSS framework lacks interpretation/visualization on what attributes in the genomic sequences associated with antigenic and high-yield phenotypes. The machine learning algorithm should automatically provide these relationships for the input sequences in the test case.

Response: Thank you for your insightful feedback. We have now launched a public webserver (<http://sysbio.missouri.edu/software/MAIVeSS>) for users to utilize. On this platform, users can input HA sequences from public datasets or their own custom sequences. In return, the model will provide either the antigenic cartography or cell/egg-based growth data. To enhance user experience, we've integrated functionalities such as zoom-in, zoom-out, dataset selection, and other features. Additionally, comprehensive instructions on how to navigate and use the webserver are provided.

3) In Ln 85, the authors claimed that the same principles can be readily applied to other subtypes of influenza viruses. How to convince the readers about the generalization ability of the proposed principles? Is there web server available to potential users which could select influenza vaccine seeds for other subtypes?

Response: Kindly refer to our previous responses. We contend that the problem formulation of sparse learning, applied to phenotype-genotype associations, can be extended to other influenza virus types, like H3N2. For such H3N2 strains, securing a high-yield growth for the vaccine remains a challenge. However, it's essential to note that the models would require training using subtype-specific datasets.

4) In the recent publication (<https://doi.org/10.1038/s41467-023-39199-6>), a machine learning-guided antigenic evolution prediction (MLAEP), which combines structure modeling, multi-task learning, and genetic algorithms to predict the viral fitness landscape and explore antigenic evolution via in silico directed evolution. What are the advantages of MAIVeSS framework in comparison with MLAEP? Moreover, did the authors explore the power of language models when they developed MAIVeSS?

Response: Thank you for referring this article, in which the authors proposed a novel machine learning model, so called Machine Learning-guided Antigenic Evolution Prediction (MLAEP), which applies transformer network to model the binding affinities between RBD and eight monoclonal RBD specific antibody and between RBD and ACE2 proteins. The genetic algorithm was used to generate synthetic RBD variants, and the binding affinities to those antibodies are evaluated by the model, which will identify those with potential mutant escape while maintaining high ACE2 binding ability. This study brought a novel concept on how to integrate both structure and sequences for machine learning.

As a comparison, different from the classification problem defined by MLAEP, MAIVeSS quantify antigenic distances between viruses with the input data of serological datasets with polyclonal sera, which are available for influenza surveillance. With the distance matrix predicted by MAIVeSS, an antigenic map was constructed to visualize their antigenic relationship. By tracking how the antigenic relationships between strains change over time, we can learn about antigenic evolution that is driving these changes.

Compared with MLAEP, MAIVeSS uses conventional machine learning for better interpretability. During the model development stage, MAIVeSS learned amino acid features associated with antigenicity using 10-fold cross-validation without involving the testing data. In contrast, MLAEP does not directly learn features during the model development stage. Instead, MLAEP identifies positions that frequently appear in the antigenic variants, which are synthetic sequences (the testing data) classified by the model. Specifically, MLAEP constructs a position frequency matrix (PFM) and calculates the Kullback-Leibler divergence (KL divergence) for each position by comparing the PFMs of the training data and synthetic data. Thus, the inherent features associated with model performance are less intuitive and interpretable in MLAEP than MAIVeSS.

MLAEP has an advantage in its utilization of a transformer network to systematically integrate structural information, which MAIVeSS currently lacks. Although we have already incorporated amino acid proximity derived from structures into our learning process, we intend to adopt a similar strategy from MLAEP in our future studies. We have expanded the “Related machine learning methods” section in our discussion in the Supplementary Information.

Unfortunately, we can not access to the training codes for this model MLAEP at Github (<https://github.com/WHan-alter/MLAEP>), which was also noted by another user through Github. Thus, we can not compare our model with MLAEP directly. We followed the reviewer’s suggestion and explored three natural language models, including Long short-term memory (LSTM), Gated recurrent units (GRU), and Transformer. We compared our method with these models (see Supplementary Tables 1-3), supporting MAIVeSS is an effective model.

Reviewer #3 (Remarks to the Author):

The authors have addressed my comments.

Reviewer #4 (Remarks to the Author):

The authors have adequately addressed the reviewer's concerns.

Reviewer #5 (Remarks to the Author):

The paper proposes a machine-learning framework focused on formulating a regularized loss function aimed at supporting optimal candidate vaccine virus (CVV) selection. I have been invited to review the R1 version of the manuscript. I am impressed by the comments raised by the reviewer of the original submission and the comprehensive response provided by the authors. It's evident that the authors have addressed all previous comments, particularly the two relevant to the methodology.

The authors have updated the detailed data splitting procedures to explain how they fine-tune parameters and compare performance. Additionally, they have shared materials (datasets, codes, and other resources) that enable researchers to delve into important details. In my independent reading, this manuscript is well-written, organized, and perfectly illustrated.

Fundamentally, I recommend accepting this paper as is.